# EF-P and its paralog EfpL (YeiP) differentially control translation of proline-containing sequences

Alina Sieber [1,8], Marina Parr [2,8], Julian von Ehr [3,4], Karthikeyan Dhamotharan [3], Pavel Kielkowski [5], Tess Brewer [1], Anna Schäpers[1], Ralph Krafczyk [1], Fei Qi [6], Andreas Schlundt [3,7], Dmitrij Frishman [2] & Jürgen Lassak [1]✉

Polyproline sequences are deleterious to cells because they stall ribosomes. In bacteria, EF-P plays an important role in overcoming such polyproline sequence-induced ribosome stalling. Additionally, numerous bacteria possess an EF-P paralog called EfpL (also known as YeiP) of unknown function. Here, we functionally and structurally characterize EfpL from *Escherichia coli* and demonstrate its role in the translational stress response. Through ribosome profiling, we analyze the EfpL arrest motif spectrum and find additional sequences beyond the canonical polyproline motifs that both EF-P and EfpL can resolve. Notably, the two factors can also induce pauses. We further report that EfpL can sense the metabolic state of the cell via lysine acylation. Overall, our work characterizes the role of EfpL in ribosome rescue at proline-containing sequences, and provides evidence that co-occurrence of EF-P and EfpL is an evolutionary driver for higher bacterial growth rates.

Decoding genetic information at the ribosome is a fundamental trait shared among all living organisms. However, translation of two or more consecutive prolines leads to ribosome arrest[1–6]. To allow translation to continue, nearly every living cell is equipped with a specialized elongation factor called e/aIF5A in eukaryotes and archaea, or EF-P in bacteria[7,8]. Upon binding close to the ribosomal tRNA exiting site (E-site), EF-P stimulates peptide bond formation by stabilizing and orienting the peptidyl-tRNA[Pro][9,10]. EF-P has a three-domain structure that spans both ribosomal subunits[10,11] and consists of an N-terminal Kyprides, Ouzounis, Woese (KOW) domain and two oligonucleotide binding (OB) domains[12], together mimicking tRNA in size and shape[13]. Although this structure is conserved among all EF-P homologs[14], bacteria have evolved highly diverse strategies to facilitate proper interactions between EF-P and the CCA end of the P-site tRNA[Pro]. For

instance, in *Escherichia coli*, a conserved lysine K34 at the tip of the loop bracketed by two beta strands β3 and β4 (β3Ωβ4) of the KOW domain is post-translationally activated by β-D-lysylation using the enzyme EpmA[15–19]. Firmicutes such as *Bacillus subtilis* elongate lysine K32 of their EF-P by 5-aminopentanolylation[20], while e.g., in β-proteobacteria or pseudomonads, an arginine is present in the equivalent position, which is α-rhamnosylated by the glycosyltransferase EarP[14,21,22]. Among the remaining EF-P subtypes the paralogous YeiP (from now on termed EfpL for "EF-P like") sticks out, as it forms a highly distinct phylogenetic branch (Fig. 1A; Supplementary Fig. 1)[14,23]. However, to date, the molecular function of EfpL remains enigmatic[24]. Bioinformatic analyses based on AlphaFold predictions indicate that EF-P-like proteins have a three-domain structure similar to EF-P, but they only share about 30% sequence similarity. Across the

[1]Faculty of Biology, Microbiology, Ludwig-Maximilians-Universität München, Planegg-Martinsried, Germany. [2]Department of Bioinformatics, Wissenschaftszentrum Weihenstephan, Technische Universität München, Freising, Germany. [3]Institute for Molecular Biosciences and Biomolecular Resonance Center (BMRZ), Goethe University Frankfurt, Frankfurt, Germany. [4]IMPRS on Cellular Biophysics, Frankfurt, Germany. [5]Department of Chemistry, Institut für Chemische Epigenetik (ICEM), Ludwig-Maximilians-Universität München, Munich, Germany. [6]State Key Laboratory of Cellular Stress Biology, School of Life Sciences, Xiamen University, Xiamen, China. [7]Institute of Biochemistry, University of Greifswald, Greifswald, Germany. [8]These authors contributed equally: Alina Sieber, Marina Parr. ✉e-mail: juergen.lassak@lmu.de

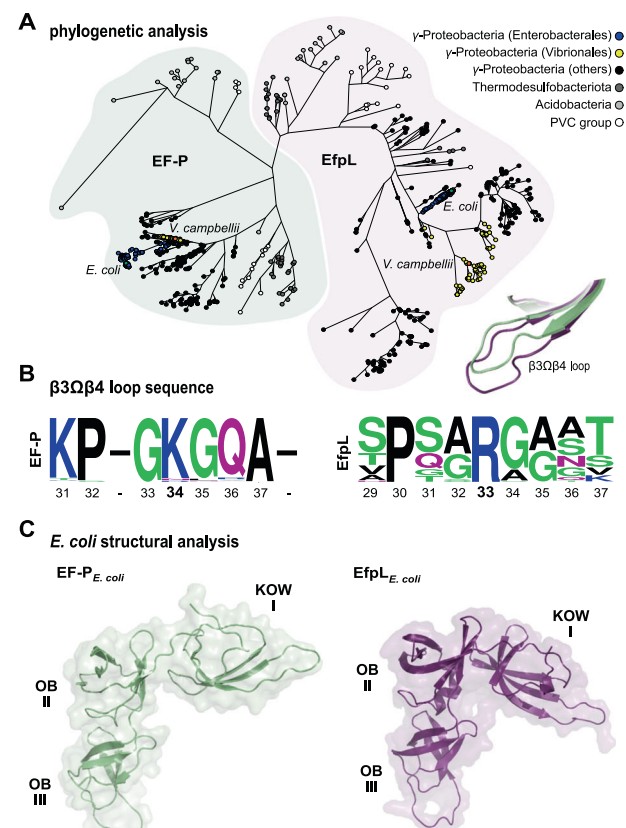

**Fig. 1 | Structural and phylogenetic analysis of the EfpL subgroup.**
**A** Phylogenetic tree of EfpL (purple) and co-occurring EF-Ps (green). Colors of tip ends depict bacterial clades. Comparison of the KOW β3Ωβ4 loop in *E. coli* EF-P (taken from PDB: 6ENU; green) and EfpL (PDB: 8S8U, this study; purple).
**B** Sequence logos[104] of β3Ωβ4 loop of EfpL and co-occurring EF-Ps. **C** Comparison of structures of *E. coli* EF-P (taken from PDB: 6ENU) and EfpL (PDB: 8S8U, chain B, this study) with overall fold views and three domains.

three domains, the C-terminal OB-domain shows the highest similarity between the two proteins. This domain's primary role is to interact with the small ribosomal subunit and the anticodon stem loop of the P-site tRNA. Notably, both EF-P and EfpL contain a tyrosine and an arginine in position 183 and 186, respectively (according to *E. coli* EF-P numbering), which are close enough to form hydrogen bonds with A42 of the P-site tRNA and G1338 within helix h29 of the 16S rRNA[10]. By contrast, the key residues in the KOW domain of EF-P, as well as the residue involved in specific recognition of prolyl-tRNA in stalled ribosomes, are less conserved. This in turn suggests that EfpL's role in translation diverges from those of canonical EF-Ps. In the frame of this study, we solve the structure of *E. coli* EfpL (EfpL) and uncover its role in translation of XP(P)X-containing proteins: Through ribosome profiling, we explore the EfpL arrest motif spectrum and uncover additional sequences beyond the typical polyproline motifs that both EF-P and EfpL can resolve. Additionally, these factors can also trigger translational pauses. Moreover, we demonstrate that EfpL is capable of detecting the cell's metabolic state via lysine acylation.

## Results

### Structural and phylogenetic analysis of EfpL revealed unique features in the β3Ωβ4 loop

We began our study by recapitulating a phylogenetic tree of EF-P in order to extract the molecular characteristics of the EfpL subgroup. A collection of 4736 complete bacterial genomes from a representative set that covers species diversity was obtained from the RefSeq database[25]. From these organisms, we extracted 5448 EF-P homologs

and identified the branch that includes the "elongation factor P-like protein" of *E. coli*. This subfamily comprises 528 sequences (Supplementary Figs. 1 and 2; Supplementary Data 1) and is characterized by a number of unique features (Fig. 1). First, we observed that EfpL is predominantly found in Proteobacteria of the γ-subdivision but also in Thermodesulfobacteria, Acidobacteria and the Planctomycetes/Verrucomicrobia/Chlamydiae-group (Fig. 1A). This suggests a similar but more specialized role in translation than EF-P. Second, we noted that the EfpL branch is most closely affiliated but still separated from the arginine-type EF-P subgroup, which is activated by α-rhamnosylation, a reaction catalyzed by the glycosyltransferase EarP (Supplementary Fig. 1)[14,21,22]. This evolutionary connection extends beyond overall sequence similarity to the functionally significant β3Ωβ4 loop (Fig. 1A) and the arginine (R33 in *E. coli* EfpL) at its tip (Fig. 1B; Supplementary Fig. 2C)[14]. However, in contrast to these α-rhamnosylated EF-Ps, R33 in EfpL remains unmodified, as confirmed by mass spectrometry (MS) (Supplementary Fig. 3). Additionally, we discovered a strictly conserved proline three amino acids upstream of EfpL_R33−an amino acid typically absent from that position in α-rhamnosylated EF-Ps[14]. Third, EfpLs predominantly co-occur with the EF-P subfamily activated by β-D-lysylation whereas the presence of an EF-P that is α-rhamnosylated typically excludes the existence of the paralogous EfpL (Supplementary Fig. 2B)[23]. Lastly, distinguishing itself from all other EF-Ps, EfpL appears to possess a β3Ωβ4 loop extension (Fig. 1A, B; Supplementary Fig. 2D). However, the exact length of this extension remains ambiguous in the in silico models.

Accordingly, we solved the crystal structure of *E. coli* EfpL (PDB: 8S8U; Supplementary Data 2A) and compared it with other available protein structures of EF-P[10,26]. This confirmed the highly conserved fold of EF-P typed proteins in prokaryotes, both expressed by a structural overlay and respective root-mean-square deviation (r.m.s.d.) values (Supplementary Fig. 4). The EfpL structure reveals a significantly tilted KOW domain relative to the C-terminal di-domain compared to EF-P structures (Fig. 1C), certainly enabled by the flexible hinge region between the independent moieties. However, a separate alignment of KOW and OB di-domains between *E. coli* EfpL and for example, the EF-P structure resolved within the *E. coli* ribosome from Huter et al.[10], reveals low r.m.s.d. values (Supplementary Fig. 4). This suggests the relative domain arrangement is merely a consequence of the unique crystal packing. Altogether, the EfpL high-resolution structure reveals the anticipated fold and features needed for its expected functional role interacting with the ribosome, analogously to EF-P. We then took a closer look at the KOW domain β3Ωβ4 loop relevant for interacting with the tRNA. The structural alignment ultimately revealed a β3Ωβ4 loop elongation by two amino acids for EfpL, different from the canonical seven amino acids in EF-P (Fig. 1A, B; Supplementary Fig. 2D). In this way, EfpL_R33 remains apical similar to canonical EF-Ps. We reasoned that such a loop extension would enable unprecedented contacts with the CCA end of the P-site tRNA without further posttranslational modification, which we set out to investigate in detail. Given the overall structural similarity with EF-P, we overlaid the EfpL KOW domain with the cryo-EM structure of EF-P bound to the ribosome[10] to analyze the position and potential contacts of EfpL_R33 with that tRNA trinucleotide. In EF-P, the modified K34 aligns with the trinucleotide backbone without obvious RNA-specific interactions, while the prolonged sidechain allows for a maximum contact site with the RNA (Supplementary Fig. 5). To allow for local adjustments in an otherwise sterically constrained frame of the ribosome, we carried out molecular docking of EfpL and the CCA trinucleotide with a local energy minimization using HADDOCK[27] (Supplementary Fig. 5; Supplementary Data 2B). As shown by the lowest-energy model, the local geometry in principle would allow the unmodified arginine of the EfpL β3Ωβ4 loop to reestablish the interaction with the tRNA trinucleotide. Furthermore, the model suggests EfpL could mediate specific interactions with the RNA as−unlike EF-P_K34−EfpL_R33 was found to stack

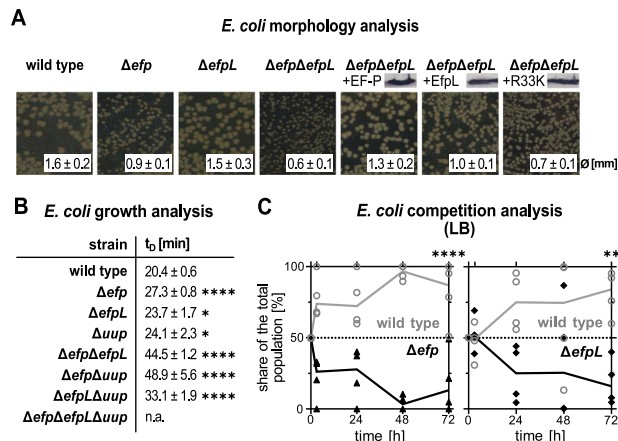

**A**

*E. coli* morphology analysis

wild type | Δ*efp* | Δ*efpL* | Δ*efp*Δ*efpL* | Δ*efp*Δ*efpL* +EF-P | Δ*efp*Δ*efpL* +EfpL | Δ*efp*Δ*efpL* +R33K

1.6 ± 0.2 | 0.9 ± 0.1 | 1.5 ± 0.3 | 0.6 ± 0.1 | 1.3 ± 0.2 | 1.0 ± 0.1 | 0.7 ± 0.1 Ø [mm]

**B** *E. coli* growth analysis

| strain | $t_D$ [min] |
|---|---|
| wild type | 20.4 ± 0.6 |
| Δ*efp* | 27.3 ± 0.8 **** |
| Δ*efpL* | 23.7 ± 1.7 * |
| Δ*uup* | 24.1 ± 2.3 * |
| Δ*efp*Δ*efpL* | 44.5 ± 1.2 **** |
| Δ*efp*Δ*uup* | 48.9 ± 5.6 **** |
| Δ*efpL*Δ*uup* | 33.1 ± 1.9 **** |
| Δ*efp*Δ*efpL*Δ*uup* | n.a. |

**C** *E. coli* competition analysis (LB)

**Fig. 2 | The role of EfpL in bacterial physiology. A** Morphology analysis of *E. coli* BW23113 and isogenic mutant strains lacking *efp* (Δ*efp*), *efpL* (Δ*efpL*), or both genes (Δ*efp*Δ*efpL*). In strains overproducing EF-P (+EF-P), EfpL (+EfpL) and EfpL_R33K (+R33K) protein production was confirmed by immunoblotting utilizing the C-terminally attached His$_6$-tag and Anti-His$_6$ antibodies (α-His). Colony size was quantified by averaging the diameters (mean Ø ± standard deviation (sd)) of 30 colonies on LB agar plates after 18 h of cultivation at 37 °C. Morphology analysis on plates was repeated two more times with similar results. **B** Doubling times (mean $t_D$ ± sd) were calculated from exponentially grown cells in LB ($n ≥ 6$, biological replicates). Statistically significant differences to wild-type growth according to two-way ANOVA test (*P* value (*P*) **P* < 0.0332, *****P* < 0.0001). **C** Growth analysis of *E. coli* cells in mixture over 72 h. An *E. coli* strain Δ*cadC* without any mutant growth phenotype under the test conditions[92] was used as wild type. *E. coli* BW25113 Δ*cadC* was mixed with either *E. coli* BW25113 Δ*efp* or Δ*efpL* and cultivated for 72 h. The share of the population was detected on LB agar plates ($n = 4$, biological replicates). Statistically significant differences to wild-type growth according to two-way ANOVA test (***P* = 0.003, *****P* < 0.0001). **A–C** Source data are provided as a Source Data file.

between the two C-bases and make polar interactions with the phosphate-sugar backbone. Hence, based on the docking model we suggest that the prolonged β3Ωβ4 loop and its central tip R33 are capable of compensating for the lack of a modified lysine. It will be interesting to see an atom-resolved proof for this interaction in future high-resolution structures that provide insight beyond the limitations of a docked model.

## *E. coli* EF-P and EfpL have overlapping functions

Based on the structural similarities (Fig. 1C), we assumed that EF-P and EfpL have a similar molecular function. However, there has been no experimental evidence supporting this hypothesis so far. Accordingly, we analyzed growth of *E. coli* wild type and mutants lacking *efp* (Δ*efp*), *efpL* (Δ*efpL*), or both genes (Δ*efp*Δ*efpL*) (Fig. 2A, B). Compared to the strong mutant phenotype in Δ*efp* ($t_d$ ~27 min), we observed a slight but still significant increase in doubling time from ~20 min in the wild type to ~24 min in Δ*efpL*. In line with this observation, a paralleling competition experiment demonstrated that wild-type cells outcompete not only Δ*efp* but also Δ*efpL* within 72 h (Fig. 2C). Further, the mild growth phenotype in Δ*efpL* becomes pronounced in the double deletion mutant Δ*efp*Δ*efpL*, which impairs growth beyond the loss of each single gene ($t_d$ ~45 min). This implies a cooperative role in the translation of polyproline proteins, which is almost masked by EF-P in Δ*efpL* cells. The overproduction of either EF-P or EfpL, but not the substitution of the functional important R33 at the β3Ωβ4 loop tip in the EfpL_R33K variant, completely or partially eliminates the growth defect. However, this effect vanishes when the functional important R33 at the β3Ωβ4 loop tip is substituted in the EfpL_R33K variant, demonstrating the significance of R33 for the molecular function of EfpL. It is also noteworthy that overproduction of EfpL in Δ*efp*Δ*efpL*

reduced doubling time below that of Δ*efp* (~27 min) (Supplementary Fig. 6). We hypothesize that ectopic expression partially compensates for the comparatively low-copy number of EfpL per cell (EfpL: ~4500 vs. EF-P: ~40,000 in complex medium[28]) (Supplementary Fig. 7A, B). A similar phenotypic pattern was observed for *efp* and *efpL* deletions when examining the same strains in terms of the CadABC-dependent pH stress response (Supplementary Fig. 8)[29], whose regulator CadC has a polyproline motif[3].

Parallel to our work another player in ribosome rescue at proline-containing arrest motifs was described: an ABCF ATPase termed Uup in *E. coli* and YfmR in *B. subtilis*[30–32]. Notably, while the phenotypic consequences of losing *yfmR* or *efp* hardly affect vegetative growth of *B. subtilis*, their simultaneous deletion dramatically impacts viability and was even suggested to be synthetically lethal. However, there is no ortholog of EfpL in *B. subtilis*. We consequently asked what happens when we delete *uup* in our previously introduced *efp* and *efpL* mutant strains (Fig. 2A). We were able to construct the two double deletions Δ*efp*Δ*uup*, and Δ*efpL*Δ*uup* but we failed to generate a triple deletion Δ*efp*Δ*efpL*Δ*uup* (Fig. 2B). This only succeeded in the presence of a plasmid-encoded, arabinose-inducible copy of *efpL* (Δ*efp*Δ*efpL*Δ*uup* +EfpL). Subsequent growth analyses confirmed that the presence of the inducer allowed *E. coli* to reach cell numbers similar to those of the wild type (and all single and double deletion strains) (Supplementary Fig. 9). By contrast, repression of *efpL* transcription reduced the viable cell counts of *E. coli* Δ*efp*Δ*efpL*Δ*uup* +EfpL by five orders of magnitude. Altogether, this led us to conclude that all three proteins have an overlapping arrest spectrum, and that EfpL becomes essential for ribosome rescue at consecutive prolines when *efp* and *uup* are absent. To confirm this latter hypothesis and pinpoint EfpL's molecular functions in relieving ribosome arrest on diprolines, we used our recently described reporter assay (Fig. 3A)[33]. This assay allows positive correlation of translational pausing strength with bioluminescence. Deletion of either *efp* or *efpL* leads to an increased light emission, and for Δ*efp*Δ*efpL*, we observed a cumulative effect. Again, the phenotype of Δ*efp*Δ*efpL* was trans-complemented by wild-type copies of the respective genes. A parallel quantitative in vitro assay employing NanoLuc® variants with and without polyproline insertion (Fig. 3B) confirmed the results of the previous in vivo experiments with EfpL and its substitution variant EfpL_R33K (Fig. 3C). Unlike in the in vivo analyses with Δ*efpL* and Δ*efp* strains, there are no significant differences in the rescue efficiency between EF-P and EfpL at the tested diproline motif PPN.

## *E. coli* EF-P and EfpL alleviate ribosome stalling at distinct XP(P)X motifs with differences in rescue efficiency

To elucidate the EfpL arrest motif spectrum, a ribosome profiling analysis (RiboSeq) was conducted. Here an *E. coli* wild type was compared with Δ*efp* and Δ*efpL* strains. Importantly, we also included Δ*efp* cells in which EfpL was overproduced. As indicated by our previous analyses (Figs. 2 and 3) this compensates for the relatively low natural copy number of the factor and might uncover motifs that are otherwise masked by the presence of EF-P. We used PausePred[34] to predict pauses in protein translation in the respective strains. Subsequently, we calculated the frequencies of amino acid triplet residues occurring at the sites of predicted pauses (Fig. 4A; Supplementary Data 3A). In line with the molecular function of EF-P, diproline motifs were heavily enriched at pause sites in Δ*efp*[3,5,6]. As already suspected by the mild mutant phenotype of the *efpL* deletion (Figs. 2 and 3) we did not see a significant difference between Δ*efpL* and the wild type. However, in stark contrast, overproduction of EfpL alleviated ribosome stalling at many but not all arrest motifs identified in Δ*efp*. Further, in line with EF-P function, our comparative metagene analysis revealed no noticeable effects on initiation or termination for EfpL (Supplementary Figs. 10 and 11)[35]. Together this corroborates the idea that EfpL has evolved to assist EF-P in translational rescue. Our analysis further

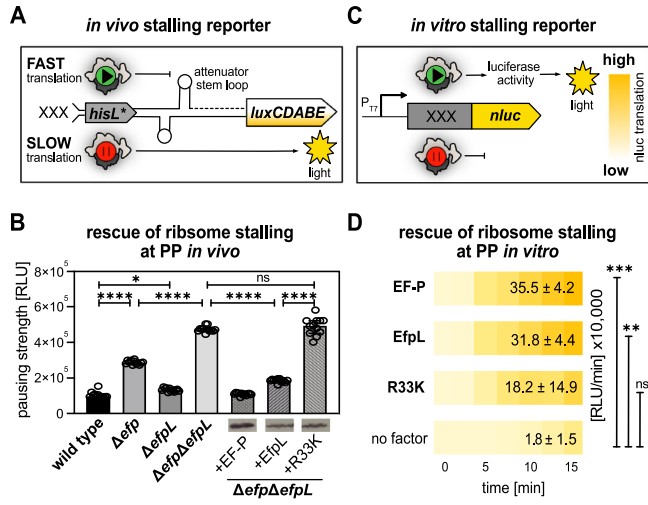

**Fig. 3 | The function of EfpL in alleviation of ribosome stalling. A** Scheme of the in vivo stalling reporter system[33]. The system operates on the histidine biosynthesis operon of *E. coli*. In its natural form, the histidine biosynthesis gene cluster is controlled by the His-leader peptide (HisL), which comprises seven consecutive histidines. In our setup, the original histidine residues (His1 through His4) were replaced by artificial sequence motifs (XXX). Non-stalling sequences promote the formation of an attenuator stem loop (upper part) that impedes transcription of the downstream genes, thus ultimately preventing light emission. Conversely, in the presence of an arrest motif, ribosomes pause and hence an alternative stem loop is formed that does not attenuate transcription of the *luxCDABE* genes of *Photo-rhabdus luminescens*. **B** In vivo comparison of pausing at PPN in *E. coli* (for strain labeling and immunoblotting details see (**A**)). Pausing strength is given in relative light units (RLU) ($n = 12$, biological replicates, mean with sd indicated as error bars). Statistically significant differences according to an ordinary one-way ANOVA (*$P < 0.0332$, ****$P < 0.0001$, ns not significant). **C** Scheme of the in vitro cell-free stalling reporter assay. The system is based on *nanoluc* luciferase (nluc®) which is preceded by an artificial sequence motif (XXX). DNA is transcribed from a T7 promoter ($P_{T7}$) using purified T7 polymerase (NEB). Pausing strength is proportional to light emission. **D** In vitro transcription and translation of the nLuc® variant nLuc_PPN. The absence (no factor) or presence of the respective translation elongation factors of *E. coli* (EF-P, EfpL, EfpL_R33K) is shown. Translational output was determined by measuring bioluminescence in a time course of 15 min and endpoints are given in relative light units (RLU/min±sd) ($n \geq 3$, technical replicates). Statistically significant differences to the control (no factor) according to ordinary one-way ANOVA (**$P = 0.0015$, ***$P = 0.0005$, ns not significant). **B**, **D** Source data are provided as a Source Data file.

revealed that among the top 29 stalling motifs are not only XPPX but also many XPX motifs and one motif completely lacking a proline (Fig. 4A; Supplementary Fig. 12). The RiboSeq findings were confirmed with our in vivo luminescence reporter (Fig. 4B; Supplementary Fig. 13) by testing 12 different arrest motifs as well as in vitro by quantifying production of two NanoLuc® Luciferase (nLuc) variants comprising IPW and PAP (Fig. 4C; Supplementary Fig. 14). Together, our data demonstrate that while a P-site proline is almost always a prerequisite for ribosome rescue by EF-P/EfpL, in rare cases motifs lacking proline can also be targeted.

An arrest spectrum extension beyond diprolines has only been reported for IF5A thus far[35,36] although there are weak indications in the literature that EF-P[4] and similarly EfpL might assist in synthesis of the XPX containing sequence of the leader peptide MgtL[37–40], which we able to substantiate (Supplementary Fig. 15). To further explore EfpL's contribution to gene-specific translational rescue, we focused on the top 29 motifs as done before for eIF5A[35] and looked at the frequency of ribosome occupancy before and after the pause sequence. The ratio between these values gives an asymmetry score (AS) and provides a good measure for stalling strength[6]. EF-P and EfpL dependency was

determined by comparing with the AS from the wild type. We were thus able to recapitulate the data from previous RiboSeq analyses for the Δ*efp* samples (Supplementary Data 3B, C). Moreover, with this approach, we were able to find EfpL targets not only in the Δ*efp* +EfpL sample but also in Δ*efpL*. In line with our phenotypic analyses (Fig. 2; Supplementary Figs. 8 and 15), most of these proteins are also targeted by EF-P (Fig. 4D; Supplementary Data 3). While in the majority of cases, the rescue efficiency was better with EF-P, we found some proteins where EfpL seems to be superior. We even identified a few candidates that were only dependent on EfpL. The proteins targeted by EfpL are frequently involved in amino acid metabolism and transport (Fig. 4D; Supplementary Data 3D). This provides a potential explanation for the growth phenotype we observed in Lysogeny broth (LB), where amino acids constitute the major source of nutrients (Fig. 2A–C). Notably, when we swapped to glucose as dominant C-source and compared growth in LB and LB supplemented with 20 mM Glucose, indeed the cumulative growth defect of Δ*efp*Δ*efpL* was gone (Supplementary Fig. 6). Moreover, while wild-type *E. coli* outcompetes Δ*efp* under these conditions, the proportion of the Δ*efpL* population remained constant within 72 h (Fig. 4E). Thus, our data support the assumption that EF-P functions as a housekeeping factor whereas EfpL exerts its role depending on the available nutrients. We hypothesize, that the structural differences between the two factors lead to different efficiencies in resolving ribosome stalling at specific motifs (Supplementary Data 3)[30–32,41]. A sequence logo based on translations modulated exclusively by EfpL (Supplementary Fig. 12C) shows a clear over-representation of DPA, PPV and DPN (Supplementary Data 3C) and, presumably depending on the amino acid context of the arrest motif[42,43], this factor will become superior in resolving the stall.

**A guanosine in the first position of the E-site codon as recognition element for EF-P and EfpL**

The chemical nature of the X residues in XP(P)X in the top 29 stalling motifs (Fig. 4A; Supplementary Fig. 12A, B) is highly diverse and does not provide a cohesive rationale for the arrest motif spectrum: besides the negatively charged residues aspartate and glutamate, we found especially the hydrophobic amino acids isoleucine and valine as well as small ones, like glycine for X at the XP(P) position (Supplementary Fig. 12C). Consequently, we extended our view to the codon level. EF-P and accordingly EfpL can interact with the E-site codon utilizing the first loop in the C-terminal OB-domain (d3 loop I)[10,11]. We did not see any preference for a specific base in the wobble position. By contrast, we revealed a strong bias for guanosine in the first position of the E-site codon in the sequence logos (Supplementary Fig. 12D) of EF-P- and EfpL-targeted XPP motifs, where X ≠ P. Notably, we observed no clear trend when we looked at the X in (P)PX in motifs (Supplementary Fig. 12C). When bound to the ribosome, EF-P establishes contacts with the first and second position of the E-site codon through d3 loop I residues G144–G148, with sidechain-to-base specific contacts involving D145 and T146[10] (Supplementary Fig. 16). However, in the available high-resolution structure, ribosomes are arrested at a triproline motif and thus, the E-site codon (CCN) does not contain a guanosine. Referring to our observation, we replaced the cytosine in the structure by guanosine in silico, followed by an additional docking and energy minimization of the loop-RNA interface (Supplementary Fig. 16A, B). The resulting complex suggests additional contacts that in principle could appear between guanosine and EF-P as compared to cytosine (Supplementary Fig. 16C). Despite potential biases of the docking procedure, a preference for G would be supported by an extended interface with sequential contacts up to residue G151. As suggested by the model, this could per se involve the entire d3 loop I. Based on the motif analysis, we thus conclude that especially guanosine in the first position of the E-site codon promotes EF-P and EfpL binding to the ribosome, which is additionally supported by the in silico comparison.

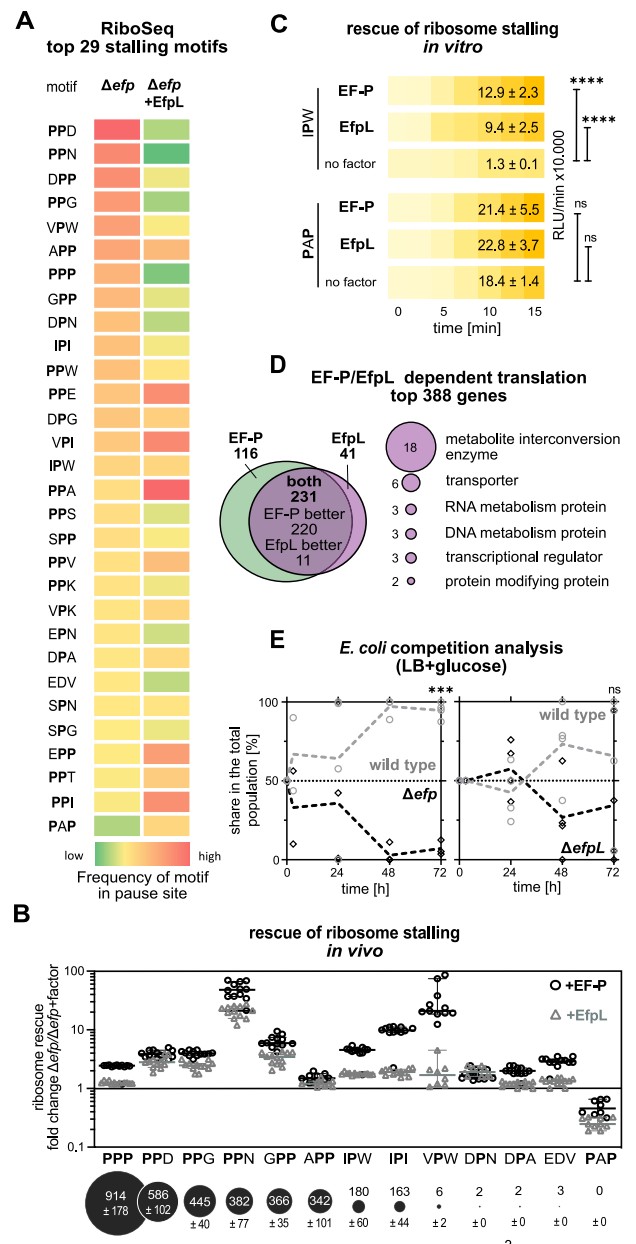

**Fig. 4 | The target spectrum of EF-P and EfpL. A** Color code of the heat map corresponds to frequency of the motif to occur in pause site in the ribosome profiling analysis predicted with PausePred[34] (From green to red = from low to high). First column: Top 29 motifs whose translation is dependent on EF-P and the control motif PAP in the ribosome profiling analysis comparing *E. coli* BW25113 with the *efp* deletion mutant (Δ*efp*). Second column: Comparison of profiling data of Δ*efp* and Δ*efp* cells overexpressing *efpL* (Δ*efp* +EfpL) at these motifs. **B** In vivo comparison of rescue efficiency of a set of stalling motifs and the control PAP. Given is the quotient of relative light units measured in Δ*efp* and corresponding trans-complementations by EF-P (+EF-P) and EfpL (+EfpL). Motifs are sorted according to pausing strength determined with our stalling reporter (*n* = 12, biological replicates, mean with sd indicated as error bars). **C** In vitro transcription and translation of *nLuc®* variants nLuc_3xRIPW (IPW) and nLuc_3xRPAP (PAP). The absence (no factor) or presence of the respective translation elongation factors of *E. coli* (EF-P/EfpL) is shown. Translational output was determined by measuring bioluminescence in a time course of 15 min and is given in relative light units measured at the end of the reaction (RLU/min ± sd) (*n* ≥ 3, technical replicates). Statistically significant differences to control (no factor) according to ordinary one-way ANOVA (****$P$ < 0.0001, ns not significant). **D** Left part: Venn diagram of top 388 genes, whose translation depends on EF-P and EfpL. Dependency was determined by comparing asymmetry scores from genes encompassing top 29 stalling motifs listed in (**A**). Right part: Enriched protein classes to which EfpL-dependent genes belong[126]. **E** Growth analysis of *E. coli* cells in mixture over 72 h in LB with 40 mM glucose. A Δ*cadC* strain without growth phenotype[92] was used as the wild type. *E. coli* BW25113 Δ*cadC* was mixed with either *E. coli* BW25113 Δ*efp* or Δ*efpL*. The share of the population was detected on LB agar plates (*n* = 4, biological replicates). Statistically significant differences to wild-type growth according to two-way ANOVA (***$P$ = 0.0006, ns not significant). **A**–**C**, **E** Source data are provided as a Source Data file.

## EF-P and EfpL can induce translational pauses

We found the unique recognition elements of an EF-P/EfpL-dependent arrest motif to be the P-site tRNA^Pro and the E-site codon, in agreement with past studies[9,10,44]. We therefore wondered whether XP—regardless of being part of a stalling motif or not—promotes binding of EF-P and similarly EfpL to the ribosome. If so, such "off-binding" might induce pausing at non-stalling motifs instead of alleviating it. Although weak, we indeed saw that loss of *efp* increases pausing with our PAP non-stalling control (Fig. 4B), which comprises two XPX motifs namely RPA and APH. Conversely, *efp* and *efpL* overexpression showed the opposite effect. Thus, our study provides evidence that the translation factors EF-P and EfpL can induce pausing, presumably by blocking tRNA translocation to the E-site. Our hypothesis was confirmed by showing that one can also induce pausing at a clean APH motif (Supplementary Fig. 17A). Either such an apparently deleterious effect is accepted, as the positive influence on arrest motifs outweighs the negative one, or translational pauses at XP(P)X might also have positive effects on, for example, buying time for domain folding or membrane insertion[45]. We were further curious whether we see codon-

specific effects and tested the non-stalling motif RPH, in which the E-site codon starts with C (R is encoded by CGC) (Supplementary Fig. 17A). Congruent with our previous findings EF-P could no longer increase pausing strength and with EfpL the effect was less pronounced, while an R33K substitution had no inhibitory effect. In summary, our findings indicate that EF-P (and EfpL) may be able to bind to the ribosome whenever a proline is translated, with binding being further promoted by the E-site codon. This idea is in line with earlier work from Mohapatra et al.[46]. The authors reported that EF-P binds to ribosomes during many or most elongation cycles. Our data may now provide a rationale for this (at the time) unexpectedly high binding frequency, which by far exceeds the number of XPPX arrest motifs. In addition to these weak pauses induced at XPX, we observed in our RiboSeq data that EF-P might also bind non-productively at certain motifs as evidenced by asymmetry scores that are higher in Δ*efp* samples than in the wild type (Supplementary Data 3B, C). While such events are predominantly weak and only rarely observed in our Δ*efp* RiboSeq data, their frequency and strength increased when we overproduced EfpL in the Δ*efp* +EfpL sample (Supplementary Fig. 17; Supplementary Data 3B, C). This supports the idea that the structural differences of the two factors differentially align and stabilize the P-site tRNA^Pro. We thus reasoned that the presence of a constitutive EF-P and a more specialized EfpL, would provide the cell with a lever to intentionally delay or accelerate translation gene specifically. However, this would require regulation. Following indications from a global analysis, *efpL* expression is regulated by carbon catabolite repression (Supplementary Fig. 18A)[47]. It was predicted that P_*efpL* is a class II cAMP response protein (CRP)-dependent promoter. However, the putative CRP binding site deviates significantly from the consensus motif of the regulator. Consequently, we reinvestigated the hypothesized regulation analogous to previous studies[48] but did not observe any measurable effect (Supplementary Fig. 18B, C). Subsequently, we extended our dataset to include conditions such as nutrient availability, acetyl-phosphate levels, heat, cold, acidic and alkaline pH, as well as high and low osmolarity (Supplementary Fig. 18D, E). Under all tested conditions, the promoter activities of P_*efp* and P_*efpL* maintained a constant

ratio of about 10:1. Our findings are also consistent with "The quantitative and condition-dependent *E. coli* proteome"[28], which shows that the protein copy number patterns of EF-P and EfpL perfectly match and follow other ribosomal factors (Supplementary Fig. 18F). Accordingly, post-transcriptional control of the respective *efp* and *efpL* mRNAs is attributed to maintaining the balance in protein copy number between the two proteins[49–52].

### *E. coli* EfpL is deactivated by acylation

As an alternative to protein copy number control, post-translational modifications provide a means to adjust EF-P activity to cellular needs. Since we were able to demonstrate that—unlike many other EF-P subtypes—the EfpL β3Ωβ4 loop tip is unmodified, we extended our view to the entire protein sequence. The idea arose as the activity and subcellular localization of the eukaryotic EF-P ortholog eIF5A is regulated by phosphorylation and acetylation, respectively[53,54]. A literature search revealed that *E. coli* EfpL is acylated at four different lysines (K23, K40, K51, and K57) in the KOW domain (Fig. 5A)[55–58]. Notably, a sequence comparison with EF-P shows that a lysine is found only in the position equivalent to K57, and there is no evidence of modification[55–58]. Possible acylations of EfpL encompass not only acetylation but also malonylation and succinylation (Fig. 5A). As a consequence, the positive charge of lysine can either be neutralized or even turned negative. To investigate the impact of acylation on EfpL we generated protein variants in which we introduced $N_\varepsilon$-acetyllysine (AcK) by amber suppression[59] at each individual position where acylation was previously reported (EfpL_K23AcK, EfpL_K40AcK, EfpL_K51AcK, and EfpL_K57AcK). Testing of purified protein variants in the established in vitro assay revealed that K51 acetylation impairs EfpL's function, significantly (Fig. 5B; Supplementary Fig. 19). We argue that charge alterations at these lysines, as well as subsequent steric constraints, will impair ribosomal interactions. To this end, we modeled the EfpL KOW domain to the ribosome by structural alignment with EF-P in order to investigate the effects of acetylation visualized by respective in silico modifications (Supplementary Fig. 20). In line with the rescue experiments, the in silico data show that compared to all other modification sites K51 is most sterically impaired by acetylation. Longer sidechain modifications at K51 such as succinylation will most likely prevent EfpL from binding to the ribosome. To confirm the in silico and in vitro data on EfpL inactivation by acylation, we sought to validate these findings in vivo. Acylation is predominantly a non-enzymatic modification influenced by the cell's metabolic state (Supplementary Fig. 21A, B), specifically by internal levels of acetyl-phosphate[55,56]. Consequently, different growth conditions can either promote or inhibit acylation levels. For instance, glucose or acetate utilization increases acetylation of EfpL due to higher levels of the acetyl group donor, acetyl phosphate, which is particularly significant for K51[55,56,60]. We used *E. coli* cells that dependent on EfpL as the sole ribosome rescue system for stalls at XPX and XPPX (Δ*efp*Δ*uup*), and tested growth in acetate medium, expressing EfpL K51 substitutions (Fig. 5C). Arginine (K51R) was used to mimic the non-acetylated state, glutamine (K51Q) served as an acetyllysine mimic, and glutamate (K51E) introduced a negative charge similar to malonylation and succinylation. All variants were expressed from a low-copy number plasmid[61] under the control of the native *efpL* promoter (P*efpL*). Under these conditions, only the K51R culture grew comparable to Δ*efp*Δ*uup* cells ectopically expressing *efp*. By contrast, the culture with the K51Q variant turned only slightly turbid and we did not observe an increase in culture density for the K51E variant nor with wild-type EfpL. This, in turn, confirmed our previous assumptions, demonstrating that both chain length and charge at EfpL position 51 are crucial for protein activity. In conclusion, our combined in vivo, in vitro, and in silico data clearly demonstrate that EfpL is inactivated by acylation. It has recently been shown that acetylation of ribosomal proteins in general inhibits translation and increases the proportions of dissociated 30S and 50S

ribosomes[62]. In addition to this scheme, we have now uncovered, that in *E. coli* the activity of EfpL is regulated by acylation. In this way, the protein acts as a sensor for the metabolic state to regulate translation of specific XP(P)X proteins.

### The presence of EfpL is associated with faster bacterial growth

Paralogous proteins evolve to diversify functionality and enable species-specific regulation[63]. In this regard, we found that in enterobacteria, the four acylation sites of EfpL in *E. coli* remain largely invariable, whereas in others, such as *Vibrio* species, they show less conservation (Fig. 5D; Supplementary Fig. 22). Most importantly, lysine in position 51 is an arginine in the EfpLs of e.g., *Vibrio cholerae, Vibrio natriegens* and *Vibrio campbellii*. Moreover, we found that expression levels of *efpL V. campbellii* (*efpL$_{Vca}$*) are much higher than in *E. coli* and equal those of *efp$_{Vca}$*, together suggesting a broader role for EfpL in this organism (Supplementary Fig. 23). We compared the rescue efficiency of EfpL$_{Vca}$ with those of selected *Enterobacteriaceae* (Fig. 5E) and found that overproduction of EfpL$_{Vca}$ was superior over all tested enterobacterial EfpLs. In fact, the protein could most effectively counteract the translational arrest at PPN not only in vitro but also in vivo (Fig. 5E, F; Supplementary Fig. 24). Next, we investigated the effect of an *efpL$_{Vca}$* deletion. Similar to *E. coli* we did not find any growth phenotype. However, in stark contrast, a deletion of *efp$_{Vca}$* also had no consequences for growth speed. Only the simultaneous deletion of both genes (Δ*efp*Δ*efpL*) diminished growth in *V. campbelli*, suggesting that EfpL$_{Vca}$ and EF-P$_{Vca}$ can fully compensate for the absence of the other. To exclude a species-specific behavior, we further included *V. natriegens*, the world record holder in growth speed (doubling time is less than 10 min under optimal conditions) (Fig. 5G)[64]. Similar to *V. campbellii* both proteins seem to be of equal importance. Therefore, we conclude, that the role of EfpL in ribosomal rescue of XP(P)X is more general in *Vibrio* species compared to Enterobacteria.

We were ultimately curious, whether there might be a universal benefit for bacteria in encoding EfpL. To this end, we estimated doubling times of a reference dataset of γ-proteobacteria using a codon usage bias-based method (Supplementary Fig. 25)[65]. Then we categorized them according to presence or absence of an EfpL paralog. To minimize differences resulting from phylogenetic diversity we focused specifically on γ-proteobacteria encoding an EF-P that is activated by EpmA (Fig. 5H). Notably, bacteria with EfpL are predicted to grow faster than those lacking it. Thus, we conclude that the concomitant presence of EF-P and EfpL might be an evolutionary driver for faster growth. We speculate that microorganisms with both proteins benefit from their unique capabilities to interact with the P-site tRNA$^{Pro}$, which in turn helps to increase overall translation efficiency.

## Discussion

Proline is the only secondary amino acid in the genetic code. The pyrrolidine ring can equip proteins with unique properties[66] and the polyproline helix is just one expression for the structural possibilities[67]. However, all this comes at a price. The rigidity of proline decelerates the peptidyl transfer reaction with tRNA$^{Pro}$. Not only is it a poor A-site peptidyl acceptor, but also proline is a poor peptidyl donor for the P-site[68,69]. Nevertheless, arrest-inducing polyprolines occur frequently in pro- and eukaryotic genomes[45,70]. This, in turn, shows that the benefits of such sequence motifs outweigh the corresponding drawbacks and explain why nature has evolved the universally conserved EF-P to assist in translation elongation at XP(P)X[33]. To promote binding to the polyproline stalled ribosome EF-P specifically interacts with the D-loop of the P-site tRNA$^{Pro9}$, the L1 stalk, and the 30S subunit[11] and the mRNA[10], with the latter being the only variable in this equation. Accordingly, in the ideal case, the EF-P retention time on the ribosome could be modulated according to the motif's arrest strength. Indeed, the dissociation rate constant of EF-P from the ribosome differs

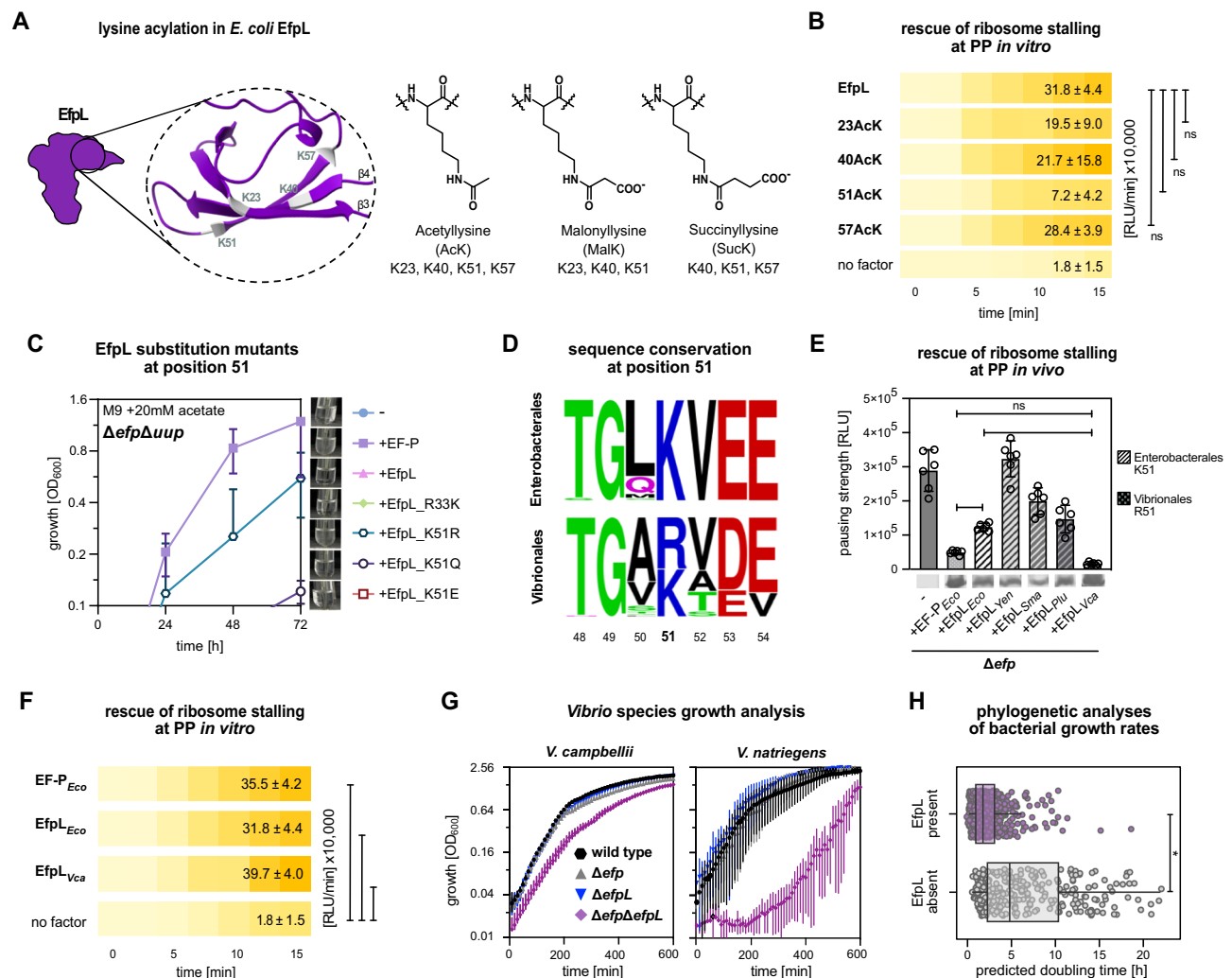

**Fig. 5 | EfpL acylation and its regulation in distinct bacteria. A** EfpL acylations according to refs. [55–58]. Acylated lysines are depicted as part of a polypeptide, represented by the wavy line. **B** In vitro transcription and translation of the *nLuc®* variant nLuc_PPN. The absence (no factor) or presence of *E. coli* EF-P or EfpL and substitution variants EfpL_K23AcK, EfpL_K40AcK, EfpL_51AcK, EfpL_K57AcK is shown. Translational output was determined by measuring bioluminescence in a 15 min time course and is given in relative light units (RLU/min±sd) ($n \geq 3$, technical replicates). Statistically significant differences according to ordinary one-way ANOVA (*$P = 0.0364$, ns not significant). **C** Growth analysis of *E. coli* BW25113 Δ*efp*Δ*uup* trans-complemented with *efp* (+EF-P), *efpL* (+EfpL) or *efpL* substitution mutants (+EfpL_R33K/_K51R/_K51Q/_K51E) in M9-medium with 20 mM acetate as sole carbon source. Images of growth media were taken after 48 h ($n = 3$, biological replicates, mean with sd indicated as error bars). **D** Sequence logos[104] of position 51 ± 3 amino acids in EfpL in Enterobacterales and Vibrionales. **E** In vivo comparison of pausing at PPN in *E. coli* Δ*efp* cells and trans-complementations with EF-P/EfpL of *E. coli* (+EF-P*Eco*/+EfpL*Eco*), *Yersinia enterocolitica* (+EfpL*Yen*), *Serratia marcescens*

(+EfpL*Sma*), *P. luminescens* (+EfpL*Plu*), *Vibrio campbellii* (+EfpL*Vca*). Pausing strength is given in relative light units (RLU) ($n = 6$, biological replicates, mean with sd indicated as error bars). Statistically significant differences according to one-way ANOVA (*$P = 0.0152$, ***$P = 0.0002$, ns not significant). **F** In vitro analysis as in (**B**). The absence (no factor) or presence of elongation factors of *E. coli* (EF-P*Eco*/EfpL*Eco*) and *V. campbellii* (EfpL*Vca*) is shown. ($n \geq 3$, technical replicates) (statistics as in (**B**), ****$P = 0.0001$, **$P = 0.0015$, ***$P = 0.0005$). **G** Growth analysis of *V. campbellii* (in LM) and *Vibrio natriegens* (in LB) with corresponding deletions of *efp* (Δ*efp*), *efpL* (Δ*efpL*), or both genes (Δ*efp*Δ*efpL*) ($n = 11$; biological replicates, mean with sd indicated as error bars). **H** Phylogenetic analysis of predicted γ-proteobacterial growth rates comparing absence or presence of EfpL. Doubling times were predicted using codon usage bias in ribosomal proteins. ($n = 786$ genomes, median with top and bottom boundaries representing 1st and 3rd quartiles and whiskers indicating 1.5 times inter-quartile range). Statistically significant difference according to phylogenetic ANOVA ($P = 0.029$, $P$ value based on 1000 permutations). **B**, **C**, **E**–**G** Source data are provided as Source Data file.

depending on the E-site codon[44]. Our data support the hypothesis that amino acids encoded by a codon beginning with a guanosine induce a particularly strong translational arrest in XP(P) motifs (Supplementary Fig. 12). As EF-P is an ancient translation factor being already present before phylogenetic separation of bacteria and eukaryotes/archaea[71], we wondered whether there is a connection to the evolution of the genetic code. Remarkably, all six amino acids encoded by GNN (Gly, Ala, Asp, Glu, Val, Leu) are included among the standard amino acids that can be produced under emulated primordial conditions[72]. One might therefore speculate that in the early phase of life, EF-P/IF5A were essential to assist in nearly every peptide bond formation with proline

in the P-site and thus reading the E-site codon by a second OB-domain was especially beneficial.

The importance to alleviate ribosome stalling at prolines is further underlined by the existence of additional rescue systems namely YebC1 and YebC2[73] (orthologous to the mitochondrial TACO1[74]) and the ATP-Binding Cassette family-F (ABCF) protein Uup in *E. coli* and its ortholog YfmR in *B. subtilis*[30–32,41,75]. In interplay with EF-P, Uup/YfmR, YebC, and EfpL can facilitate translation of XP(P)X-containing proteins. The different modes of action and structural characteristics of the four factors enabled specialization. In case of EfpL, the protein is superior in ribosome rescue at specific genes (Fig. 4D; Supplementary Data 3).

This, in turn, might be an evolutionary driving force for translational speed and hence higher growth rates as indicated by our phylogenetic analysis (Fig. 5H). Alternatively, an EF-P paralog opens additional regulatory possibilities. In contrast to EfpL-encoding bacteria, some lactobacilli, for instance, have two copies of *efp* in the genome (Supplementary Data 1)[14]. One might speculate that here one *efp* is constitutively expressed and the second copy is transcriptionally regulated according to the translational needs. Although relying only on one EF-P, such regulation was reported for Actinobacteria, in which polyproline-containing proteins are concentrated in the accessory proteome[76]. Here EF-P accumulates during early stationary phase and might boost secondary metabolite production as evidenced for *Streptomyces coelicolor*. By contrast, for *E. coli* EfpL there is no evidence for such copy number control, as it simply mirrors the expression pattern of other ribosomal proteins[28]. Instead, the protein seems to fulfill a dual role in this organism. On the one hand, it is essential for full growth speed (Figs. 2A and 5H). On the other hand, it acts as sensor of the metabolic state (Supplementary Fig. 6). The combination of multiple sites of acylation[55–58] and the chemical diversity of this modification type[60] lead to a highly heterogenic EfpL population, which could fine-tune translation in each cell differently. We speculate that regulation of translation by acylation[62] in general and of EfpL in particular adds to phenotypic heterogeneity and thus might contribute to survival of a population under changing environmental conditions[77]. Such a scenario is particularly important for bacteria that colonize very different ecological niches, such as many enterobacteria including *E. coli* do. Depending on whether they are found e.g., in the soil/water or in the large intestine, the nutrient sources they rely on change. Therefore, it is plausible to assume that fine-tuning metabolic responses by acylating and deacylating, EfpL gives enterobacteria an advantage to thrive in the gastrointestinal tract.

Compared to the eukaryotic and archaeal IF5A, EF-P diversity is much greater[8,78]. Especially the functionally significant β3Ωβ4 has undergone significant changes. Starting with the catalytic residue at the loop tip, which is not restricted to lysine as for eukaryotes/archaea. Instead, one also finds asparagine, glutamine, methionine, serine and glycine, besides arginine[23]. These changes extend to the overall sequence composition of β3Ωβ4 to either increase stiffness[79] or, in the case of EfpL, to prolong the loop, as shown in this study by an EfpL high-resolution structure. The latter two strategies functionalize the protein without modification. Notably, the EfpL subgroup is phylogenetically linked most closely to the EF-P branch being activated by α-rhamnosylation[14,21]. This raises the question about the evolutionary origin of EfpL. Starting from a lysine-type EF-P[71], we speculate that upon gene duplication and sequence diversification, an early EfpL arose, and cells benefitted from improved functionality in a subset of XP(P)X arrest peptides. Further evolutionary events could include the shrinkage of the loop back to the canonical seven amino acids and eventually the phylogenetic recruitment of EarP. Such phylogenetic order is supported by an invariant proline upstream of the catalytically active loop tip residue which is found in EfpLs and lysine-type EF-Ps, but is absent in EarP-type EF-Ps (Fig. 1).

Lastly, EF-P diversity holds also potential for synthetic biology applications. Reportedly, EF-P can boost peptide bond formation with many non-canonical amino acids (ncAA)[80–83]. This includes not only proline derivatives but also D- and β-amino acids. However, in all studies, *E. coli* EF-P was used. Given the structural differences between EfpL and EF-P and the resulting differences in the rescue spectrum, we speculate that use of EfpL might be especially beneficial for genetic code expansion for certain ncAA. Collectively, our structural and functional characterization of the EfpL subfamily not only underscores the importance of ribosome rescue at XP(P)X motifs but also adds another weapon to the bacterial arsenal for coping with this type of translational stress. We further illustrate how different bacteria utilize this weapon to gain evolutionary advantages and give an outlook on how EfpL can potentially be used as a molecular tool.

## Methods

### Plasmid and strain construction

All strains, plasmids, and oligonucleotides used in this study are listed and described in Supplementary data files (Supplementary Data. 4), respectively. Kits and enzymes were used according to manufacturer's instructions. Plasmid DNA was isolated using the Zyppy® Plasmid Miniprep Kit from Zymo Research. DNA fragments were purified from agarose gels using the Zymoclean® Gel DNA Recovery Kit or from PCR reactions using the DNA Clean & Concentrator®-5 DNA kit from Zymo Research. All restriction enzymes, DNA modifying enzymes, and the Q5® high fidelity DNA polymerase for PCR amplification were purchased from New England BioLabs.

Plasmids for expression of C-terminally His$_6$-tagged *efp* and *efpL* genes under the control of an inducible promoter were generated by amplification of the corresponding genes from genomic DNA using specific primers and subsequent cut/ligation into the pBAD33 vector[84]. Plasmids for expression of SUMO-tagged *efpL* genes were generated with the Champion™ pET-SUMO Expression System from Invitrogen™ according to manufacturer's instructions. HisL*_lux reporter strains were generated according to Krafczyk et al.[33]. Briefly, an upstream fragment (containing desired mutations) of the *hisLGDCBHAFI* operon was amplified via PCR using the respective primer pairs. After purification, these fragments were isolated from an agarose gel, digested with specific restriction enzymes, and then ligated into the suicide vector. The resulting plasmids were introduced into *E. coli* BW25113 by conjugative mating with *E. coli* WM3064 as the donor strain on LB medium supplemented with meso-α,ε-Diaminopimelic acid (DAP). Single-crossover integration mutants were selected on LB plates containing kanamycin and lacking DAP. Finally, the resulting mutations were confirmed by Sanger sequencing.

Deletions and chromosomal integrations of His$_6$-tagged encoding genes using RecA-mediated homologous recombination with pNPTS138-R6KT of *efp* and *efpL* were made according to Lassak et al.[85,86]. To achieve this, ~500 bp long upstream and downstream fragments of the desired gene region were amplified via PCR using the respective primer pairs. After purification, these fragments were combined through overlap PCR. The final product was isolated from an agarose gel, digested with specific restriction enzymes, and then ligated into the suicide vector. The resulting plasmids were introduced into *E. coli* BW25113 or *Vibrio* species by conjugative mating with *E. coli* WM3064 as the donor strain on LB medium supplemented with DAP. Single-crossover integration mutants were selected on LB plates containing kanamycin and lacking DAP. Single colonies were cultured overnight in LB without antibiotics and subsequently plated onto LB containing 10% (wt/vol) sucrose to select for plasmid excision. Kanamycin-sensitive colonies were screened for targeted deletions through sequencing using primers flanking the site of mutation.

Genetic manipulations via Red®/ET® recombination were done with the Quick & Easy *E. coli* Gene Deletion Kit (Gene Bridges, Heidelberg, Germany). Reporter plasmid constructions with pBBR1-MCS5-TT-RBS-lux were made according to Gödeke et al.[87]. Briefly, the upstream region of genes of interest was cloned 5′ to the *luxCDABE* operon.

### Growth conditions

*E. coli* cells were routinely grown in Miller-modified LB[88,89], super optimal broth (SOB)[90] or M9 minimal medium supplemented with 20 mM of Glucose[86] at 37 °C aerobically under agitation unless indicated otherwise. *V. campbellii* cells were grown in Luria marine (LM) medium (LB supplemented with an additional 10 g/l NaCl)[91] at 30 °C aerobically. *V. natriegens* cells were grown in LB at 30 °C aerobically. Growth was recorded by measuring the optical density at a wavelength

of 600 nm ($OD_{600}$). When required 1.5% (w/v) agar was used to solidify media. Alternative carbon sources and media supplements were added and are indicated. If needed, antibiotics were added at the following concentrations: 100 μg/ml carbenicillin sodium salt, 50 μg/ml kanamycin sulfate, 20 μg/ml gentamycin sulfate, 30 μg/ml chloramphenicol. Plasmids carrying pBAD[84] or Lac promoter were induced with L(+)-arabinose at a final concentration of 0.2% (w/v) or Isopropyl-β-D-thiogalactopyranosid (IPTG) at a final concentration of 1 mM, respectively.

## In vivo promotor activity assay

*E. coli* cells harboring the plasmids pBBR1-MCS5-P*efp*-*luxCDABE* or pBBR1-MCS5-P*efpL*-*luxCDABE* versions were inoculated in LB with appropriate antibiotics. The next day, 96-well microtiter plates with fresh LB with supplements or M9 minimal media with mentioned carbon sources and were inoculated with the cells at an $OD_{600}$ of 0.01. The cells were grown aerobically in the CLARIOstar® PLUS at 37 °C, 25 °C or 42 °C. $OD_{600}$ and luminescence were recorded in 10 min intervals over the course of 16 h. Light emission was normalized to $OD_{600}$. Each measurement was performed in triplicates as a minimum.

## LDC assay

Cells were cultivated in LDC indicator medium (indicator: bromothymol blue) for 16 h and the pH increase was shown qualitatively as a color change[3].

## MgtL reporter assay

*E. coli* cells harboring the plasmids pBBR1-MCS5-*mgtL_luxCDABE* were inoculated in M9 minimal supplemented with the appropriate antibiotics and grown aerobically at 37 °C. The next day, a microtiter plate with fresh M9 minimal medium initially leaving out $Mg^{2+}$ ($Mg^{2+}$-free M9). Indicated concentrations of $Mg^{2+}$ (added as $MgSO_4$) were added subsequently. Cells were inoculated with a starting $OD_{600}$ of 0.01. Then cells were grown aerobically in the CLARIOstar® PLUS at 37 °C. $OD_{600}$ and luminescence was recorded in 10 min intervals over the course of 16 h. Light emission was normalized to $OD_{600}$. Each measurement was performed in triplicates as a minimum.

## Measurement of pausing strength in vivo

The pausing strength of different motifs was determined according to Krafczyk et al.[33] by measuring absorption at 600 nm (Number of flashes: 10; Settle time: 50 ms) and luminescence emission (Attenuation: none; Settle time: 50 ms; Integration time: 200 ms) with a Tecan Infinity® or ClarioStar plate reader in between 10-min cycles of agitation (orbital, 180 rpm, amplitude: 3 mm) for around 16 h.

## Competition experiments

For a direct comparison of *E. coli* lacking either *efp* or *efpL* with *E. coli* expressing both, different mixtures of *E. coli* BW25113 strains were analyzed over a time-course experiment. An *E. coli* strain Δ*cadC* without any mutant growth phenotype under the test conditions[92] was used as wild type. Single strains were incubated overnight at 37 °C shaking, washed in LB the following day, and resuspended to an $OD_{600}$ of 1. *E. coli* BW25113 Δ*cadC* was mixed with either *E. coli* JW4107-1 (BW25113 Δ*efp*::*Kan^R*) or JW5362-1 (BW25113 Δ*efpL*::*Kan^R*) and *E. coli* JW4094-5 (BW25113 Δ*cadC*::*Kan^R*) was mixed with either *E. coli* BW25113 Δ*efp* or BW25113 Δ*efpL*, to a starting $OD_{600}$ of 0.01 and cultivated for 3 h in LB or LB with 20 mM glucose. 100 cells from each mixture were plated on LB and LB with 50 μg/ml Kanamycin agar plates, respectively. Cultures were diluted to an $OD_{600}$ of 0.001 and cultivated for 24 h. On the next day colonies on the plates were counted, and the share of the population was calculated. The process was repeated as necessary.

## Protein overproduction and purification

For in vitro studies, C-terminally $His_6$-tagged EF-P and EfpL variants were overproduced in *E. coli* LMG194 harboring the corresponding pBAD33 plasmid. C-terminally $His_6$-tagged EfpL with acetyllysine instead of lysine at position 23, 40, 51 or 57 were overexpressed from pBAD33_*efpL*K23Amber_$His_6$, pBAD33_*efpL*K40Amber_$His_6$, pBAD33_*efpL*K51Amber_$His_6$, or pBAD33_*efpL*K57Amber_$His_6$ in *E. coli* LMG194 which contained the additional plasmid pACycDuet_AcKRST[59]. This allowed for amber suppression utilizing the acetyllysine-tRNA synthetase (AcKRS) in conjunction with PylT-tRNA. LB was supplemented with 5 mM $N^\epsilon$-acetyl-L-lysine and 1 mM nicotinamide to prevent deacetylation by CobB[93]. During exponential growth, 0.2% (w/v) L(+)-arabinose was added to induce gene expression from pBAD vectors, and 1 mm IPTG served to induce gene expression of the pACycDuet-based system. Cells were grown overnight at 18 °C and harvested by centrifugation on the next day. The resulting pellet was resuspended in HEPES buffer (50 mM HEPES, 100 mM NaCl, 50 mM KCl, 10 mM $MgCl_2$, 5% (w/v) glycerol, pH 7.0). Cells were then lysed using a continuous-flow cabinet from Constant Systems Ltd (Daventry, UK) at 1.35 kbar. The resulting lysates were clarified by centrifugation at 4 °C at 234 998 × *g* for 1 h. The $His_6$-tagged proteins were purified using Ni-NTA beads (Qiagen, Hilden, Germany) according to the manufacturer's instructions, using 20 mM imidazole for washing and 250 mM imidazole for elution. In the final step, the purified protein was dialyzed overnight against HEPES buffer to remove imidazole from the eluate.

For MS analysis cells with chromosomally encoded $His_6$-tagged EfpL were grown in SOB until mid-exponential growth phase and harvested by centrifugation. To overproduce EfpL proteins LMG194 harboring a pBAD33 plasmid with C-terminally $His_6$-tagged EfpL were grown in SOB and supplemented with 0.2% (w/v) L(+)-arabinose during exponential growth phase ($OD_{600}$). Cells were grown overnight at 18 °C and harvested by centrifugation on the next day. Pellets were resuspended in 0.1 M sodium phosphate buffer, pH 7.6. Cells were then lysed using a continuous-flow cabinet from Constant Systems Ltd. (Daventry, UK) at 1.35 kbar. The resulting lysates were clarified by centrifugation at 4 °C at 234 998 × *g* for 1 h. The $His_6$-tagged proteins were purified using Ni-NTA beads (Qiagen, Hilden, Germany) according to the manufacturer's instructions. For washing and elution, a gradient of imidazole (10, 25, 50, 75, 100, 150, 200, 250 mM) was used. The purified protein was dialyzed overnight against in 0.1 M sodium phosphate buffer, pH 7.6 to remove imidazole from the eluate.

For crystallization, *E. coli* BL21 cells harboring a pET-SUMO plasmid were grown in SOB and supplemented with 1 mM IPTG during the exponential growth phase. Cells were grown overnight at 18 °C and harvested by centrifugation on the next. Pellets were resuspended in 0.5 M Tris-HCl buffer, pH 7.0. Cells were then lysed using a continuous-flow cabinet from Constant Systems Ltd. (Daventry, UK) at 1.35 kbar. The resulting lysates were clarified by centrifugation at 4 °C at 234 998 × *g* for 1 h. The $His_6$-tagged proteins were purified using Ni-NTA beads (Qiagen, Hilden, Germany) according to the manufacturer's instructions, using 20 mM imidazole for washing and 250 mM imidazole for elution. The purified protein was dialyzed overnight against 0.5 M Tris-HCl buffer, pH 7.0 to remove imidazole from the eluate. 0.33 mg SUMO-protease per 1 mg protein were added and incubated overnight at 4 °C. SUMO-protease and SUMO-tag were captured using Ni-NTA beads (Qiagen, Hilden, Germany) according to the manufacturer's instructions. The protein was additionally purified via size exclusion chromatography on a Superdex 75 10/300 Increase column (Cytiva) in 20 mM Tris-HCl, 50 mM NaCl and 1 mM DTT at pH 8.0. Fractions with the protein of interest were concentrated and further subjected to anion exchange chromatography on a Resource Q (Bio-Rad) 6 ml-column to remove remaining contaminants with a NaCl salt gradient from 50 to 500 mM. The protein eluted at ~200 mM NaCl. The final sample was buffer-adjusted to 50 mM NaCl for crystallization.

## SDS−PAGE and western blotting

For protein analyses cells were subjected to 12.5% (w/v) sodium dodecyl sulfate (SDS) polyacrylamide gel electrophoresis (PAGE)[94]. To visualize proteins by UV light 2,2,2-trichloroethanol was added to the polyacrylamide gels[95]. Subsequently, the proteins were transferred onto nitrocellulose membranes, which were then subjected to immunoblotting. In a first step, the membranes were incubated either with 0.1 μg/ml anti-6×His® antibody (Abcam) or 0.1 μg/ml anti-acetylated-lysine (SIGMA). These primary antibodies, produced in rabbit, were targeted with 0.2 μg/ml anti-rabbit alkaline phosphatase-conjugated secondary antibody (Rockland) or 0.1 μg/ml anti-rabbit IgG (IRDye® 680RD) (donkey) antibodies (Abcam). Anti-rabbit alkaline phosphatase-conjugated secondary antibody was detected by adding development solution [50 mM sodium carbonate buffer, pH 9.5, 0.01% (w/v) p-nitro blue tetrazolium chloride (NBT), and 0.045% (w/v) 5-bromo-4-chloro-3-indolyl-phosphate (BCIP)]. Anti-rabbit IgG was visualized via Odyssey® CLx Imaging System (LI-COR, Inc).

## In vitro transcription/translation assay

The PURExpress In Vitro Protein Synthesis Kit from New England Biolabs was used according to the manufacturer's instructions, but reactions were supplemented with EF-P or EfpL, respectively, and a plasmid coding for *nluc* variants (Supplementary Data 4). Luminescence was measured over time. For a 12.5 μl reaction mixture, 5 μl of PURExpress solution A and 3.75 μl of solution B, 0.25 μl of Murine RNAse inhibitor (New England Biolabs), 5 μM EF-P or EfpL, and 1 ng pET16b_nluc variants are incubated under agitation (300 rpm) at 37 °C. At various time points, a 1 μl aliquot was quenched with 1 μl of 50 mg/ml kanamycin and stored on ice. Afterward, 2 μl of Nano-Glo Luciferase Assay Reagent (Promega) and 18 μl ddH$_2$O were added to induce luminescence development, which was detected by the Infinite F500 microplate reader (Tecan®). At least three independent replicates were analyzed, and the statistical significance of the result was determined using GraphPad prism.

## Ribosome profiling

*E. coli* strains BW25113, BW25113 Δ*efpL*, BW25113 Δ*efp* and BW25113 Δ*efp* complemented with pBAD33-*efpL*_His6 (+EfpL) were cultivated in LB or LB supplemented with 30 μg/mL chloramphenicol and 0.2% L-(+)-arabinose at 37 °C under aerobic conditions. Stranded mRNA-seq and ribosome profiling (RiboSeq) libraries were generated by EIRNA Bio (https://eirnabio.com) from stab cultures. *E. coli* strains were grown in 400 mL LB at 37 °C to an OD600 of 0.4. Cells were harvested from 200 mL of culture by rapid filtration through a Kontes 90 mm filtration apparatus with 0.45 μm nitrocellulose filters (Whatman). Cells were scraped from the filter in two aliquots (90% for RiboSeq/10% for RNA-seq) before being immediately frozen in liquid nitrogen. Total RNA was extracted from RNA-seq aliquots in trizol before mRNA was rRNA depleted, fractionated, and converted into Illumina-compatible cDNA libraries. RiboSeq aliquots were lysed in 600 μl ice-cold polysome lysis buffer (20 mM Tris pH 8; 150 mM MgCl$_2$; 100 mM NH$_4$Cl; 5 mM CaCl$_2$; 0.4% Triton X-100; 0.1% NP-40; 20 U/ml Superase*In; 25 U/mM Turbo DNase) by bead beating in a FastPrep-24 with CoolPrep Adapter−3 rounds at 6 m/s for 30 s in 2 mL cryovials containing 0.1 mm silica beads. Lysates were clarified by centrifugation at 10,000 × *g* for 5 min at 4 °C. Ribosomes for subsequently pelleted from lysates by ultracentrifugation at 370,000 × *g* for 1 h at 4 °C and resuspended in polysome digestion buffer (20 mM Tris pH 8; 15 mM MgCl$_2$; 100 mM NH$_4$Cl; 5 mM CaCl$_2$). Samples were then digested with 750 U MNase for 1 h at 25 °C and the reaction was stopped by adding EGTA to a final concentration of 6 mM. Following RNA purification and size selection of ribosome-protected mRNA fragments between 20 and 40 nt in length on 15% urea PAGE gels, contaminating rRNA was depleted from samples using EIRNA Bio's custom biotinylated rRNA depletion oligos for *E. coli* before the enriched fragments were converted into Illumina-compatible cDNA libraries.

Both stranded mRNA-seq libraries and RiboSeq libraries were sequenced in three replicates on Illumina's Nova-seq 6000 platform in 150PE mode to depths of 10 million and 30 million raw read pairs per sample respectively.

The sequence structure of the RiboSeq reads was as follows:

QQQ−rpf sequence−NNNNN−BBBBB−AGATCGGAAGAGCACACGTCTGAA

, where Q = Untemplated Addition, rpf sequence = the sequence of the read, N = UMI, a 5 nt are unique molecular identifiers (UMIs), B = Barcode, used to demultiplex (the fastq files have already been demultiplexed) and AGATCGGAAGAGCACACGTCTGAA is the sequence of the adapter. Cutadapt[96] was used with parameters -u 3 and -a AGATCGGAAGAGCACACGTCTGAA to remove untemplated addition and linker sequence. Untrimmed reads and those shorter than 30 nt after trimming were discarded. Next, the UMI and Barcode were removed and the UMI was used to remove duplicate sequences using a custom Python script. Both the RiboSeq and RNA-seq reads were next mapped to rRNA and tRNA sequences using Bowtie version 1.2[97]. Five RiboSeq samples were sequenced with two sequencing runs. These samples (WT_Rep3, DELTAefpL_Rep2, DELTAefpL_Rep3, DELTAefp_Rep3, and DELTAefp_plus_efpL_Rep1) were concatenated at this stage. Next, the reads were aligned to BW25113 *E. coli* genome (RefSeq accession number NZ_CP009273.1) with Bowtie using parameters (-m 1 -l 25 -n 2 -S). BAM file containing read alignments are available at the SRA archive (ID PRJNA1092679).

The A-site offset in the RiboSeq reads was estimated to be 11 nucleotides upstream of the 3′ of the mapped reads. For both RiboSeq and RNAseq reads this "A-site" position was used to indicate the genomic location of reads. Pause prediction was carried out on all RiboSeq samples using PausePred[34] with a minimum fold-change for a pause score set at 20 within two sliding window sizes of 1000 nt with a minimum coverage of 5% in the window. The analysis was carried out on aggregated alignment files that included all replicates for each strain. The frequencies of occurrence of trimers of amino acid residues at the locations identified to be pauses were calculated for all possible trimers of amino acid residues. For each trimer of amino acid residues, its frequency to be covered by the ribosome in the pause sites was calculated and normalized by dividing by the averaged frequency of the corresponding trimer to occur in the whole ribosome-protected fragments.

## Sequence data and domain analysis

HMMER v.3.4 was used to search for Pfam[98] domains "EFP_N" (KOW-like domain, PF08207.12), "EFP" (OB-domain, PF01132.20), and "Elong-fact-P_C" (C-terminal, PF09285.11) in the protein sequences of 5257 complete representative or reference bacterial genomes (RefSeq)[25]. We identified 5448 proteins from 4736 genome assemblies that contained all three domains mentioned above (e-value cutoff 0.001) and no other PFAM domains. Sequences of "EFP_N" domains from these proteins were multiply aligned using Clustal Omega v.1.2.4[99] with all default parameters, shown in a multiple sequence alignment (MSA1) (Supplementary Data 5). Phylogenetic tree (Extended Data Fig. 1) was inferred by IQ-TREE 2.0.7[100] with branch support analysis performed in ultrafast mode[101] using 1000 bootstrap alignments. LG+R8 was chosen to be the best-fit model[102] for the tree. The phylogenetic tree in Newick format is available in the Supplementary Materials (Supplementary Data 6). The MSA region comprising positions 40−52 corresponds to the β3Ωβ4 loop region KPGKGQA of the EF-P protein from *E. coli* str. K-12 substr. MG1655 (accession number NP_418571.1)[22]. The sequence of the EfpL protein (NP_416676.4) from *E. coli* str. K-12 substr. MG1655 has an extended β3Ωβ4 loop SPTARGAAT with the R residue at the tip. The phylogenetic tree was annotated according to the length of the β3Ωβ4 loop and the nature of the residue at the tip of the β3Ωβ4 loop.

Those 528 sequences that have an extended β3Ωβ4 loop of more than 7 residues and R at the tip of it formed one branch in the phylogenetic tree. Among the sequences belonging to this branch 474 are annotated as "EfpL" or "YeiP" (synonym of EfpL) proteins in the RefSeq database and no other sequences from the list (Supplementary Data 1) have this annotation. Sequences with an extended β3Ωβ4 loop of more than seven residues and the R residue at the tip of it are referred to as EfpL. The remaining 4920 sequences constituted the set of EF-P sequences. The dataset covers 4777 genomes: 4111 of them contain only one sequence with the three domains mentioned above, 660 genomes contain two such sequences, and 6 genomes—three such sequences (Supplementary Data 1). In a separate analysis step, Clustal Omega v.1.2.4[99] with all default parameters were used to multiply align the sequences of KOW-like domains of the EfpL and EF-P proteins from the EfpL-containing genomes (MSA2) (Supplementary Data 5). IQ-TREE 2.0.7[100] with LG+G4 found to be the best-fit model[102] was used to build a phylogenetic tree (Fig. 1A). The branch support analysis in ultrafast mode[103] was performed using 1000 bootstrap alignments. The phylogenetic tree in Newick format, including bootstrap values, can be found in Supplementary Data (Supplementary Data 6). We used the ggtree R package[103] to visualize the phylogenetic trees and annotate them. Sequence logos were built using Weblogo[104].

EF-P-containing genomes were scanned for the EpmA, EarP and YmfI proteins. EpmA and EarP proteins were defined as single-domain proteins containing the "tRNA-synt 2" (PF00152.20) and "EarP" (PF10093.9)[14] domains, respectively. Using HMMER v.3.4 searches we identified these proteins in 1230 and 565 genomes, respectively. Orthologs of the YmfI protein (UniProt ID: O31767) from *Bacillus subtilis*[20,78] were obtained using the procedure described in Brewer and Wagner[23]. Briefly, this involved BLASTP(ref) searches using the *B. subtilis* YmfI as the query sequence, followed by manual bitscore cutoff determination due to the homology of this protein to other broadly conserved proteins.

## EfpL structure determination

Initial crystallization trials were performed in 96-well SWISSCI plates at a protein concentration of 4.8 mg/ml using the C3 ShotGun (SG1) crystallization screen (Molecular Dimension). Rod-shaped crystals grew after 7 days at 293 K. Diffracting crystals were obtained in 100 mM Sodium-HEPES, 20% (w/v) PEG 8000 and 10 mM Hexaamminecobalt (III) chloride conditions. The crystals were cryoprotected in mother liquor supplemented with 20% (v/v) glycerol and snap-frozen at 100 K. Datasets from cryo-cooled crystals were collected at EMBL P13 beamlines at the PETRA III storage ring of the DESY synchrotron[105]. The crystals belonged to space group P 1 21 1, with unit cell dimensions of $a = 60.71$, $b = 53.46$, and $c = 64.95$ Å. Preprocessed unmerged datasets from autoproc+STARANISO[106] were further processed in CCP4cloud[107]. Phases were obtained from molecular replacement using the AlphaFold2 model[102,108] deposited under ID AF-P0A6N8-F1. The structure was built using the automatic model building pipeline ModelCraft[109], optimized using PDB-REDO[110], refined in REFMAC5[111], and BUSTER[112] with manual corrections in Coot[113]. The quality of the built model was validated with the MolProbity server[114]. The final model was visualized in PyMOL version 2.55 (Delano Scientific). The asymmetric unit contained two molecules of EfpL. In chain A, residues 145–149 in OB-III loop showed weaker electron density and thus were not built. For depiction and comparison to other structures as well as for the HADDOCK procedure, chain B was chosen based on completeness and quality of the model. The data collection and refinement statistics are shown in Supplementary Data (Supplementary Data 2A).

## Docking and modeling of EF-P and EfpL complexes

For the comparative analysis of EF-P with the E-site codons CCG or GCG through a loop in its C-terminal OB-domain, we used the available PDB entry 6ENU[10] as a starting structure. In accordance with prior definitions by the authors, we directly analyzed and visualized available contacts for the EF-P d3 loop1 around the conserved motif $_{144}GDT_{146}$ with the present $_{-3}CCG_{-1}$ trinucleotide of the peptidyl-tRNA$^{Pro}$. For contacts with a putative GCG, we initially replaced the initial C nucleotide by G in silico using PyMol (Delano Scientific) and monitored the novel contacts using the implemented tools. For a more thorough analysis, we extracted both the GCG trinucleotide and EF-P from the structure and used the two components for an in silico docking followed by energy minimization using HADDOCK[27]. Here, we defined protein residues 146, 147, and 151 as active granting full flexibility to the structure and using automated secondary structure recognition and retainment. RNA residue G-3 was defined as active to enable seed contacts. From a total of 116 structures used by for clustering by HADDOCK 49 were found in the best-scoring cluster 1 (Supplementary Data 2B). Because of very low remaining restraint violation energies, we integrated the best four models to create an average structure used to analyze contacts between EF-P and RNA.

To analyze and compare interactions of EfpL and EF-P KOW domains with the P-site codon CCA through the β3Ωβ4 loop we looked at the available contacts of the loop as given in the PDB entry 6ENU[10]. For a model of EfpL with the trinucleotide, we aligned the EfpL KOW domain as found in our crystal structure with EF-P from PDB entry 6ENU[10]. We extracted the $_{74}CCA_{76}$ trinucleotide from the latter and used the two components as starting structures for a docking and energy minimization procedure as described above. Nucleotides 74 and 75 were defined as active, and KOW domain residues 30–35 were set as fully flexible with R33 defined as explicitly active. One hundred and ninety-eight out of the 200 structures provided by HADDOCK were found in the same cluster with no measurable violations (Supplementary Data 2B).

For all HADDOCK runs, we implemented the following settings and restraints in context of the spatial and energetic constraints of the natural ribosome environment: Protein N- and C-termini were kept uncharged and no phosphates were left at nucleic acid termini. No particular RNA structure restraints have been applied and only polar hydrogens were installed in both components. For the 0th iteration, components were kept at their original positions for an initial energy-minimizing docking step. No random exclusion of ambiguous restraints was included during docking. Passive residues were defined automatically from the non-active ones using a surface distance threshold of 6.5 Å. We used a minimum percentage of relative solvent accessibility of 15 to consider a residue as accessible. In all runs 1000 initial structures were used in rigid body docking over five trials (excluding 180°-rotations of the ligand), from which the best 200 were subjected to an energy minimization step including short molecular dynamics simulations in explicit water. Default settings were used in advanced sampling parameters of the it1 and final solvated steps (Kyte−Doolittle), respectively. Standard HADDOCK settings were applied for clustering of the 200 final structures with a minimum cluster size of 4.

For the in silico analysis of modified lysines, respective sidechains were acetylated based on the EfpL crystal structure using PyMol with no further adjustments of rotamers. The modified KOW domain was then structurally aligned with EF-P in PDB entry 6ENU[10].

## Mass spectrometry for identification of modification status

For top-down EfpL measurements, the proteins were desalted on the ZipTip with C4 resin (Millipore, ZTC04S096) and eluted with 50% (v/v) acetonitrile 0.1% (v/v) formic acid (FA) buffer resulting in ~10 µM final protein concentration in 200−400 µl total volume. MS measurements were performed on an Orbitrap Eclipse Tribrid Mass Spectrometer (Thermo Fisher Scientific) via direct injection, a HESI-Spray source (Thermo Fisher Scientific) and FAIMS interface (Thermo Fisher

Scientific) in a positive, peptide mode. Typically, the FAIMS compensation voltage (CV) was optimized by a continuous scan. The most intense signal was usually obtained at −7 CV. We measured multiple spectra from the same protein sample. The MS spectra were acquired with at least 120,000 FWHM, AGC target 100 and 2–5 microscans. The spectra were deconvoluted in Freestyle 1.8 SP2 (Thermo Fisher Scientific) using the Xtract Deconvolution algorithm.

## Predicted growth rates

We used a set of 871 genomes from the class γ-proteobacteria from the Integrated Microbial Genomes database[115]. These genomes were selected to maximize diversity by including only one genome per Average Nucleotide Identity cluster. We used CheckM[116] v1.0.12 to assess the quality of each genome and retained only those that were predicted to be at least 90% complete and contain less than 5% contamination. We re-assigned taxonomy using the Genome Taxonomy Database and GTDB-Tool kit (GTDB-Tk)[117] version 0.2.2 and removed genomes where the user-reported species did not agree with GTDB (removed 2 genomes). For example, we removed a genome with a user-reported species of *Serratia marcescens 1822* which was sorted to the genus *Rouxiella* by GTDB-Tk. We also removed 14 genomes of endosymbionts from consideration, mainly from the genus *Buchnera*.

We further subset for only those genomes which contained both genes for *epmA* and *epmB* (removed 62 genomes), contained at least one *efp* gene (removed 2 genomes) and had predicted doubling times under 24 h (removed 15 genomes). This left 786 genomes for our analysis. We identified the genes for epmA, epmB, efp, and efpL (yeiP) using a combination of different functional databases. We identified *epmA* and *epmB* by searching for the COG[118] function ids COG2269 and COG1509, respectively. We identified *efp* by searching for the Pfam[119] domain pfam01132. We identified the gene for *efpL* (*yeiP*) by searching for the TIGRfam[120] annotation TIGR02178. Next, we estimated the doubling time associated with each remaining genome using the R package gRodon[65] version 1.8.0. gRodon estimates doubling times using codon usage bias in ribosomal proteins. We used phylogenetic ANOVAs to test differences in predicted doubling times between genomes that encode EfpL and those that don't. Specifically, we used the phylANOVA function from the R package phytools[121] version 2.0.3, with p values based on 1000 permutations. We made the phylogenetic tree required for this function using 43 concatenated conserved marker genes generated by CheckM. We aligned these sequences using MUSCLE[122] v3.8.1551 and built the phylogenetic tree using IQ-TREE[123] v1.6.12. We used the model finder feature[124] included in IQ-TREE to determine the best-fit substitution model for our tree (which was the LG+R10 model). For this section, we performed all statistical analyses and plotting in R version 4.3.2 and created plots using ggplot2[125] version 3.4.4.

## Reporting summary

Further information on research design is available in the Nature Portfolio Reporting Summary linked to this article.

## Data availability

The crystal structure of EfpL_{E. coli} generated in this study have been deposited in the PDB database under accession code 8S8U. The structure of EF-P_{E. coli} from Huter et al.[10] was taken from the PDB database under accession code 6ENU. The ribosome profiling data generated in this study are available at SRD ID PRJNA1092679. Data on the acylation status of EfpL under the tested conditions can be found in the following publications by Kuhn et al.[55], Weinert et al.[56], Weinert et al.[57], and Qian et al.[58]. Quantitative *E. coli* proteome analysis data of Schmidt et al.[28] was used to compare protein concentrations in different conditions. Source data are provided with this paper.

## Code availability

R scripts and all files needed to reproduce the analyses on predicted growth rates are available at: https://github.com/tessbrewer/EfpL. An archived version of this repository has been generated and is accessible via Zenodo: https://doi.org/10.5281/zenodo.13897372.

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

## Acknowledgements

We thank Kirsten Jung, Kerstin Lassak and Wolfram Volkwein for fruitful discussions and constructive criticism. We also thank Giovanni Gallo, Anja Michl, Nina Kim Hartmann, Bhavna Menon, Irem Niran Cagil and Sabine Peschek for technical assistance. Moreover, Jürgen Lassak is grateful for DFG grant LA 3658/5-1. This work was supported by Liebig fellowship from VCI to Pavel Kielkowski. Fei Qi is supported by the National Natural Science Foundation of China (grant No. 32000462). Andreas Schlundt acknowledges support through funds SCHL2062/2-1 and 2-2 from the German Research Council (DFG) and the Johanna Quandt Young Academy at Goethe (grant number 2019/AS01). Access to synchrotron beamtime at DESY Hamburg was enabled by the Block Allocation Group grant MX939.

## Author contributions

A. Sieber, R.K., A. Schäpers, and J.L. constructed strains and plasmids. A. Sieber and R.K. performed the biochemical in vivo/in vitro characterization of EfpL. A. Schäpers performed qPCR experiments. A. Sieber purified all proteins used for in vitro assays, MS analysis and X-ray crystallography. MS experiments and analysis were done by P.K. The crystallization screen was set up by J.v.E., K.D., and A. Schlundt; and J.v.E., K.D., and A. Schlundt solved the crystal structure of EfpL$_{E.\ coli}$. A. Schlundt performed in silico interaction analyses. All bioinformatic analyses were performed by M.P., T.B., F.Q., and D.F. Ribosome profiling analyses were done by M.P. and D.F. M.P. and D.F. performed phylogenetic analyses of the EF-P subgroups and T.B. performed phylogenetic analyses of bacterial growth rates. The study was designed by J.L. with contributions from R.K. and D.F. The manuscript was written by A. Sieber, M.P., and J.L. with contributions from A. Schlundt, T.B. and D.F.

## Funding

## Competing interests

The authors declare no competing interest.
