## [Peer Review File · Nature Communications]

REVIEWER COMMENTS

Reviewer #1 (Remarks to the Author):

Sieber and colleagues characterise the biological function of EfpL (YeiP), the second EF-P orthologue in *E. coli*. The authors use a combination of wet and dry techniques, combining microbiology, molecular biology and structural biology. This is an excellent comprehensive study that sheds light on the molecular function of a previously unexplored player in bacterial protein synthesis. I have only minor suggestions.

p. 2, line 31, Introduction: e/aIF-5A should be e/aIF5A. In general introduction could be expanded.

p. 2, line 34, Introduction: KOW, OB - please define these.

p. 2, line 37, 49, Introduction: 25% of all bacteria, 10% of bacteria - I would avoid these. What does it mean, % of bacterial species? How is the full set of bacterial diversity defined? Do we know how many species are there? ... and then, Gammas are massively overrepresented amongst the sequenced species. Instead, I would specify which groups of bacterial have which modification.

p. 2, line 50, Results: 'A collection of 4736 complete bacterial genomes' maybe specify that this is a representative set covering the diversity; could be just 4736. *E. coli*.

p. 2, line 62, Results: EarP-type EF-Ps. The term sounds confusing, since here we refer to modification, not necessarily to EF-P itself. Maybe just say EarP-modified? Or specify the modification?

Figure 1A: the tree needs bootstraps. Right now it is unclear how confident it is, it could be just a guide tree. Also, is proper to provide (deposit) the alignments that are used to build a tree so that one can reproduce the analysis – the tree solution can depend on the trimming.

Figure 2A: Maybe serial dilutions would make the growth defect more visible. Maybe worth showing in SI?

Figure 3A, RiboSeq: is there an effect on initiation or/and termination? Would be nice to see a metagene analysis, SI or main.

p. 7, translational stalling caused by EF-P and EfpL overexpression: what about the R33K substituted variants?

p. 10, line 345: EfpL is an evolutionary driver of faster growth. This is too strong; consider 'The presence of EfpL is associated with faster growth' or something along the lines (correlation, no directionality implied).

Reviewer #2 (Remarks to the Author):

This study investigates the role of EfpL, an EF-P paralog. EF-P is known to help ribosomes overcome stalling at polyproline sequences, which reduce translation rate. The research reveals that EfpL also assists in resolving ribosome stalls, not only at polyproline sequences (XPPX) but also at single-proline sequences (XPX). EfpL can detect the cell's metabolic state through lysine acylation. The findings suggest that EfpL is a critical factor in ribosome rescue and that its presence alongside EF-P may enhance bacterial growth rates.

The manuscript was written strong in its structural study, highlighting the structural features of EFPL and in presenting evolutionary data, supporting the idea that EFPL-containing bacteria favor faster growth. However, it is relatively weak in explaining the physiological relevance of the presence of EFPL or the regulation of EFPL, which needs to be further explained or shown experimentally.

1) In the ribosome stalling experiments, the distinguishing features of EFPL and EFP are not well highlighted, making it unclear to me the biological/physiological significance of why bacteria have both Efp and EFPL, despite the interesting phylogenetic conservation and structural differences with EFP. If EFPL supports faster growth, what is the advantage/disadvantages for strains that have both EFPL and EFP compared to those that have only one type? It is not clear what the advantage/disadvantages is of having both. Under what conditions does EFPL become important in bacteria?

2) Similarly, the biological relevance of EFPL is unclear due to its low copy number and the indistinct ribosome stalling rescue phenotype or growth phenotype. Are there any conditions where EFPL become more important than EFP? Beyond ectopic expression, in a biological context, when does the expression level or activity of EFPL's mRNA/protein replace that of EFP? If post-translational modification by acetylation is important for activity, under those conditions, for example, acetyl CoA-limiting conditions, does EFPL's ribosome stalling rescue activity replace that of EFP?

3) Related to 2), authors mentioned that the EFP/EFPL ratio is 10:1 under all experimental conditions. Is there any specific conditions that EFPL's mRNA/protein/activity increase? In an acetyl CoA-limiting condition possible?

4) EFPL acylation: Alanine has a relatively small side chain, so if K51 is substituted with the non-acetylated form N or the acetylation mimic Q, what would happen to ribosome stalling and the growth phenotype? I hope the physiological relevance of EFPL acylation regulation becomes clearer.

5) In the authors' mass spectrometry data, have you ever detected acetylation at the K51 position? For example, in glucose media? Is there any chance acetylation affects not only activity but also protein stability? Similarly, authors predicted that in glucose media, acylation will occur due to changes in acetyl CoA levels, and a difference in growth rate under these conditions was shown. Does EFPL acetylation actually occur in cells grown in glucose media? Is there any relative mRNA/protein/activity of EFPL under glucose conditions?

6) What can be concluded about the EFPL-dependent ribosome stalling motif? The conclusions of the motif analysis are unclear. For example, is there a common ribosome stalling motif among the 18 metabolic interconversion enzymes that are EFPL-dependent? How consistent is this with the *efpL*-dependent motif analyzed earlier?

7) Since authors used gene deletion instead of a substitution mutant, can you really say that the lack of difference in growth between the wild-type and *efpL* mutant in glucose is due to EFPL K51 acetylation?

Coreviewer's comments

The manuscript by Sieber et al. identifies new features of EF-P paralog based on in silico methods. The authors propose new sites where EfpL engages, beyond the well-known XPPX motifs, and describe how its activity is regulated in *E. coli*. They also showed that the guanosine in the first

position of the E-site codon is important for EF-P and EfpL recognition. The extended but unmodified $\beta 3\Omega\beta 4$ loop, conserved in EfpL, interacts with mRNA and rescues the ribosome from XP(P)X stalling sequences. EfpL is involved in promoting faster bacterial growth and sensing the metabolic state to regulate translation.

The evidence presented strongly supports the proposed model, and the manuscript is generally well-written and understandable. However, there are still some comments and suggestions for improvement:

Major comments:

As EfpL and EF-P have overlapping functions, does the expression level of efpL increase in the Δefp strain?

Lines 314-316: Why were $\Delta efpL$ mutant strains used although acylation occurs in the K51 of EfpL?

Figure 4C: Why does the doubling time of the Efp/EfpL double-deleted strain compensated with EfpL ($\Delta Efp\Delta efpL + EfpL$) decrease in glucose media even though EfpL is deactivated by the acylation? Also, as the doubling time of $\Delta efpL$ and $\Delta efp\Delta efpL$ strains decreases in glucose media, it seems there are some unpredicted effects on growth when efpL is deleted in the presence of glucose.

Minor comments:

Figures 2A and 4C: Adding growth curves will help clarify the differences between the strains (for example, in Fig 4G).

Line 741: There are two periods at the end of the sentence.

Reviewer #3 (Remarks to the Author):

"I co-reviewed this manuscript with one of the reviewers who provided the listed reports. This is part of the Nature Communications initiative to facilitate training in peer review and to provide appropriate recognition for Early Career Researchers who co-review manuscripts."

Reviewer #4 (Remarks to the Author):

This manuscript describes the results of an extensive series of studies the authors have done to elucidate the structure, function, and physiological role of EfpL, a protein found in bacteria that mitigates the pausing that occurs when ribosomes translate sequences that include runs of prolines. EfpL turns out to be a variant of EF-P, which has a broadly similar function, but EfpL's functional properties are distinctively different. Their mms represents a significant step forward in our understanding of how the elongation phase of protein synthesis is managed in bacteria.

Specific Comments:

1. The Introduction is terse, and, quite surprisingly, it includes little, if any, mention of the prior literature on EfpL, a.k.a. YeiP. There is some, and it must be discussed in the Introduction.
2. The crystallography done has some shortcomings. For one thing, $1/\sigma$ at the limiting resolution is reported to be 2.55, which is high. Traditionally, crystallographers have included data out to $1/\sigma = 2$, and papers published in the last decade or so have made a strong case for including data out to $1/\sigma = 1$. By omitting these high resolution data (or not collecting them) the authors have left a lot of quality on the table. It is also the case that for their structure, Rfree is much higher than R than it should be, which suggests that the structure reported is not fully refined. A bit more work might have yielded something better. However, some might argue that the structure obtained is good enough for the authors' purpose.
3. Figures 1E and 3F show computed structures for the interactions between external loops of EfpL with the CCA end of tRNA and anticodon loops. They are unconvincing. Extended Data Fig. 8 includes images of the electron density for both loops. For both, it is of poor quality, as the density of external loops often is due to local disorder. The authors would be wise to omit both Fig. 1E and Fig. 3F from their mms, and delete the corresponding text. They don't add much to the argument the authors are trying to make, and they could be just plain, flat out wrong.
4. There are two chains in the asymmetric unit of the crystals the authors solved. What is the relationship between the two? A cursory glance suggests that they are related by a non-crystallographic twofold. Which chain did the authors use for their structure/function analyses they did?

Thank you very much for evaluating our manuscript and providing us with the opportunity to make constructive revisions. Enclosed, you will find our detailed, point-by-point response to the reviewers' comments.

REVIEWER COMMENTS

Reviewer #1 (Remarks to the Author):

Sieber and colleagues characterise the biological function of EfpL (YeiP), the second EF-P orthologue in *E. coli*. The authors use a combination of wet and dry techniques, combining microbiology, molecular biology and structural biology. This is an excellent comprehensive study that sheds light on the molecular function of a previously unexplored player in bacterial protein synthesis. I have only minor suggestions.

We would like to sincerely thank the reviewer for their evaluation of our work and the constructive suggestions.

p. 2, line 31, Introduction: e/aIF-5A should be e/aIF5A.

We have changed e/aIF-5A into e/aIF5A.

In general introduction could be expanded.

We agree and added the following paragraph:

*“However, to date the molecular function of EfpL remains enigmatic²⁴. Bioinformatic analyses based on AlphaFold predictions indicate that EF-P-like proteins have a three-domain structure similar to EF-P, but they only share about 30% sequence similarity. Across the three domains, the C-terminal OB-domain shows the highest similarity between the two proteins. This domain's primary role is to interact with the small ribosomal subunit and the anticodon stem loop of the P-site tRNA. Notably, both EF-P and EfpL contain a tyrosine and an arginine in position 183 and 186, respectively (according to *E. coli* EF-P numbering), which are close enough to form hydrogen bonds with A42 of the P-site tRNA and G1338 within helix h29 of the 16S rRNA10. By contrast, the key residues in the KOW domain of EF-P, as well as the residue involved in specific recognition of prolyl-tRNA in stalled ribosomes, are less conserved. This in turn suggest that EfpL's role in translation diverges from those of canonical EF-Ps. In the frame of this study, we solved the structure of *E. coli* EfpL (EfpL) and uncovered its role in translation of XP(P)X-containing proteins.”*

p. 2, line 34, Introduction: KOW, OB - please define these.

We have defined KOW and OB as Kyprides, Ouzounis, Woese (KOW) domain oligonucleotide binding (OB) domain, respectively.

p. 2, line 37, 49, Introduction: 25% of all bacteria, 10% of bacteria - I would avoid these. What does it mean, % of bacterial species?

We have eliminated these concrete percentages from the text.

How is the full set of bacterial diversity defined? Do we know how many species are there? ... and then, Gammas are massively overrepresented amongst the sequenced species. Instead, I would specify which groups of bacterial have which modification.

We thank for this important hint and have rephrased the paragraph accordingly.

p. 2, line 50, Results: 'A collection of 4736 complete bacterial genomes' maybe specify that this is a representative set covering the diversity; could be just 4736. *E. coli*.

We completely agree with the reviewer and have adjusted the text:

"A collection of 4736 complete bacterial genomes from a representative set that covers species diversity was obtained from the RefSeq database²⁵.

p. 2, line 62, Results: EarP-type EF-Ps. The term sounds confusing, since here we refer to modification, not necessarily to EF-P itself. Maybe just say EarP-modified? Or specify the modification?

We changed into "α-rhamnosylated EF-Ps"

Figure 1A: the tree needs bootstraps. Right now it is unclear how confident it is, it could be just a guide tree. Also, is proper to provide (deposit) the alignments that are used to build a tree so that one can reproduce the analysis – the tree solution can depend on the trimming.

We updated our tree and included bootstrap values as suggested and color coded them (Extended Data Fig. 1 & Supplementary Fig. S1E)

Figure 2A: Maybe serial dilutions would make the growth defect more visible. Maybe worth showing in SI?

To highlight the growth defect more clearly, we included a competition experiment, now presented as Fig. 2C. Additionally, a paralleling study identified a new player in ribosome rescue at proline-containing arrest motifs: an ABCF ATPase termed Uup in *E. coli* and YfmR in *B. subtilis*. We decided to examine the growth consequences of additionally deleting uup, now shown in Fig. 2B and Supplementary Fig S6. Our results demonstrate that Uup, like EF-P, can compensate for the absence of efpL. Furthermore, efpL becomes essential when the other compensatory players are absent.

*"Parallel to our work a new player in ribosome rescue at proline-containing arrest motifs was described: an ABCF ATPase termed Uup in *E. coli* and YfmR in *B. subtilis*³¹⁻³³. Notably, while the phenotypic consequences of losing yfmR or efp hardly affect vegetative growth, their simultaneous deletion dramatically impacts viability and was even suggested to be synthetically lethal. However, there is no ortholog of EfpL in *B. subtilis*. We consequently asked what happens when we delete uup in *E. coli* our previously introduced efp and efpL mutant strains (Fig. 2A). We were able to construct the two double deletions ΔefpΔuup, and ΔefpLΔuup but we failed to generate a triple deletion ΔefpΔefpLΔuup (Fig. 2B). This only succeeded in the presence of a plasmid-encoded, arabinose-inducible copy of efpL (ΔefpΔefpLΔuup +EfpL). Subsequent growth analyses confirmed that the presence of the inducer allowed *E. coli* to reach cell numbers similar to those of the wild type (and all single and double deletion strains). By contrast, repression of efpL transcription reduced the viable cell counts of *E. coli* ΔefpΔefpLΔuup +EfpL by five orders of magnitude (Supplementary Fig. S6). Altogether, this led us to conclude that all three proteins have an overlapping arrest spectrum, and that EfpL becomes essential for ribosome rescue at consecutive prolines when efp and uup are absent.*

Figure 3A, RiboSeq: is there an effect on initiation or/and termination? Would be nice to see a metagene analysis, SI or main.

We did not see any of these effects and added the following text and figures:

"However, in stark contrast, overproduction of EfpL alleviated ribosome stalling at many but not all arrest motifs identified in Δefp. Further in line with EF-P function our comparative metagene analysis revealed no noticeable effects on initiation or termination for EfpL (Supplementary Fig. S7, S8)³⁷."

p. 7, translational stalling caused by EF-P and EfpL overexpression: what about the R33K substituted variants?

With the EfpL R33K variant the inhibitory effect of EfpL vanished. We included the data by extending Fig. S11.

"Congruent with our previous findings EF-P could no longer increase pausing strength and with EfpL the effect was less pronounced, while an R33K substitution had no inhibitory effect."

p. 10, line 345: EfpL is an evolutionary driver of faster growth. This is too strong; consider 'The presence of EfpL is associated with faster growth' or something along the lines (correlation, no directionality implied).

We adjusted the subtitle "*The presence of EfpL is associated with faster growth*" and weakened the conclusion of the chapter.

"Thus, we conclude that the concomitant presence of EF-P and EfpL might be an evolutionary driver for faster growth. We speculate that microorganisms with both proteins benefit from their unique capabilities to interact with the P-site tRNA^{Pro}, which in turn helps to increase overall translation efficiency."

Reviewer #2 (Remarks to the Author):

This study investigates the role of EfpL, an EF-P paralog. EF-P is known to help ribosomes overcome stalling at polyproline sequences, which reduce translation rate. The research reveals that EfpL also assists in resolving ribosome stalls, not only at polyproline sequences (XPPX) but also at single-proline sequences (XPX). EfpL can detect the cell's metabolic state through lysine acylation. The findings suggest that EfpL is a critical factor in ribosome rescue and that its presence alongside EF-P may enhance bacterial growth rates.

The manuscript was written strong in its structural study, highlighting the structural features of EFPL and in presenting evolutionary data, supporting the idea that EFPL-containing bacteria favor faster growth. However, it is relatively weak in explaining the physiological relevance of the presence of EFPL or the regulation of EFPL, which needs to be further explained or shown experimentally.

We thank the reviewer for their evaluation of our study. We have added additional data (see point-by-point reply below) and provided more detailed explanations of our results where necessary.

1) In the ribosome stalling experiments, the distinguishing features of EFPL and EFP are not well highlighted, making it unclear to me the biological/physiological significance of why bacteria have both Efp and EFPL, despite the interesting phylogenetic conservation and structural differences with EFP. If EFPL supports faster growth, what are the advantages/disadvantages for strains that have both EFPL and EFP compared to those that have only one type?

There are two advantages to having both EF-P and EfpL compared to bacteria that only have EF-P. Although EF-P and EfpL (Figure 2) have overlapping functions, the structural differences between EfpL and EF-P allow for different efficiencies in resolving ribosome stalling at specific motifs. This hypothesis is further supported by the discovery of a novel ribosome rescue system—termed Uup in *E. coli*—which also alleviates stalling at proline-containing arrest motifs. This discovery coincided with our analyses. To incorporate this finding, we included corresponding mutant analyses in the revised version (Figure 2B). Additionally, we supplemented the manuscript with a sequence logo

based on the translations influenced exclusively by EfpL (Extended Data Fig. 4C) and updated the text accordingly.

"Thus, our data supports the assumption that while EF-P functions as a housekeeping factor, EfpL exerts its role depending on the available nutrients. We hypothesize, that the structural differences between the two factors lead to different efficiencies in resolving ribosome stalling at specific motifs (Supplementary Tab. S3)^{31-33,43}. A sequence logo based on translations modulated exclusively by EfpL (Extended Data Fig. 4C) shows a clear overrepresentation of DPA, PPV and DPN (Supplementary Tab. S3C) and, presumably depending on the amino acid context of the arrest motif^{44,45}, will become superior in resolving the stall."

It is not clear what the advantage/ disadvantages is of having both.

We hope this question has been answered by our preceding response.

Under what conditions does EFPL become important in bacteria?

The importance of EfpL depends on the source of nutrients. To highlight this, we have restructured the manuscript and extended the original text in the revised version. We also added additional data to emphasize the significance of EfpL in relation to different nutrient sources (Fig. 2C & Fig. 3E).

*"This provides a potential explanation for the growth phenotype we observed in Lysogeny broth (LB), where amino acids constitute the major source of nutrients (Fig. 2A-C). Notably, when we swapped to glucose as dominant C-source and compared growth in LB and LB supplemented with 20mM Glucose, indeed the cumulative growth defect of $\Delta\text{efp}\Delta\text{efpL}$ was gone (Supplementary Fig. S3). Moreover, while wild-type *E. coli* outcompetes Δefp under these conditions, the proportion of the ΔefpL population remained constant within 72h (Fig. 3E). Thus, our data supports the assumption that EF-P functions as a housekeeping factor whereas EfpL exerts its role depending on the available nutrients. We hypothesize, that the structural differences between the two factors lead to different efficiencies in resolving ribosome stalling at specific motifs (Supplementary Tab. S3)^{31-33,43}. A sequence logo based on translations modulated exclusively by EfpL (Extended Data Fig. 4C) shows a clear overrepresentation of DPA, PPV and DPN (Supplementary Tab. S3C) and, presumably depending on the amino acid context of the arrest motif^{44,45}, will become superior in resolving the stall."*

2) Similarly, the biological relevance of EFPL is unclear due to its low copy number and the indistinct ribosome stalling rescue phenotype or growth phenotype.

The efpL phenotype is masked by the presence of EF-P and/or Uup. To demonstrate this, we have included analyses of uup deletion mutants in the revised version (Fig. 2B). We would also like to note that even in *E. coli*, the EfpL copy number is not particularly low (1000-2500 per cell), though it is lower compared to EF-P (new Supplementary Fig. S12). Additionally, our data from *Vibrio* EfpL shows that the factor diversifies its physiological significance from organism to organism (Fig. 4D-G & Supplementary Fig. S13).

Are there any conditions where EFPL become more important than EFP?

This is unexpected given the phylogeny of the two proteins. Our previous research (Volkwein et al. 2019) demonstrated that lysine-type EF-Ps emerged first in evolution, while arginine-type EF-Ps, including the EfpL branch, arose later. Therefore, EF-P acts as a housekeeping factor, functioning as a "generic tool," while EfpL evolved for more specialized purposes and assists EF-P. As shown by our Riboprofiling analysis, EfpL is superior in resolving stalls at certain target genes, but it will presumably never outweigh the importance of EF-P (Supplementary Tab. S3C,D).

Beyond ectopic expression, in a biological context, when does the expression level or activity of EFPL's mRNA/protein replace that of EFP?

While we observed equal expression levels of EF-P and EfpL in *Vibrio* (Supplementary Fig. S13), we did not find any condition in which *E. coli* EfpL's mRNA/protein replaces that of EF-P (Supplementary Fig. S12). The literature suggested that *efpL* expression is controlled by catabolite repression; however, we tested and falsified this hypothesis in the revised manuscript.

"Following indications from a global analysis, efpL expression is regulated by carbon catabolite repression (Supplementary Fig. S12)⁴⁹. It was predicted that P_{efpL} is a class II cAMP response protein (CRP)-dependent promoter. However, the putative CRP binding site deviates significantly from the consensus motif of the regulator. Consequently, we reinvestigated the hypothesized regulation analogous to previous studies⁵⁰ but did not observe any measurable effect (Supplementary Fig. S12B, C)."

Second, EfpL mRNA is a model transcript to analyze post-transcriptional regulation. However, referring to that publications EF-P's mRNA is virtually identical regulated:

*"Subsequently, we extended our dataset to include conditions such as nutrient availability, acetyl-phosphate levels, heat, cold, acidic and alkaline pH, as well as high and low osmolarity (Supplementary Fig. S12D,E). Under all tested conditions, the promoter activities of P_{efp} and P_{efpL} maintained a constant ratio of 10:1. Our findings are also consistent with "The quantitative and condition-dependent *Escherichia coli* proteome"²⁹, which shows that the protein copy number patterns of EF-P and EfpL perfectly match and follow other ribosomal factors (Supplementary Fig. S12F). Accordingly, post-transcriptional control of the respective *efp* and *efpL* mRNAs is attributed to maintaining the balance in protein copy number between the two proteins⁵²⁻⁵⁵."*

If post-translational modification by acetylation is important for activity, under those conditions, for example, acetyl CoA-limiting conditions, does EFPL's ribosome stalling rescue activity replace that of EFP?

EfpL acylation inactivates EfpL as shown by our *in silico* data and *in vitro* translation assays with acetylated EfpL variants (Fig. 4B). We would like to further point out that EfpL acetylation and especially the one at K51 depends on acetyl-phosphate as shown by Weinert *et al.* 2013 and Kuhn *et al.* 2014. We have added this information to the revised manuscript:

"Acylation is predominantly a non-enzymatic modification influenced by the cell's metabolic state, specifically by internal levels of acetyl-phosphate^{57,58}."

3) Related to 2), authors mentioned that the EFP/EFPL ratio is 10:1 under all experimental conditions. Is there any specific conditions that EFPL's mRNA/protein/activity increase? In an acetyl CoA-limiting condition possible?

As mentioned before the *efpL* phenotype is masked by the presence of EF-P and/or Uup. To demonstrate this, we have included analyses of *uup* deletion mutants in the revised version (Fig. 2B). We would also like to note that even in *E. coli*, the EfpL copy number is not particularly low (1000-2500 per cell), though it is lower compared to EF-P (new Supplementary Fig. S12F). Additionally, our data from *Vibrio* EfpL shows that the factor diversifies its physiological significance from organism to organism Fig. 4D-G.

4) EFPL acylation: Alanine has a relatively small side chain

We agree that alanine has a small side chain. Accordingly, we have never tested any K to A substitution variants. Instead, all variants we tested *in vitro* contained an acetylated lysine (AcK),

which we incorporated using amber suppression, as outlined in our previous work (Volkwein et al. 2019).

... so if K51 is substituted with the non-acetylated form N or the acetylation mimic Q, what would happen to ribosome stalling and the growth phenotype? I hope the physiological relevance of EFPL acylation regulation becomes clearer."

Following this great suggestion, we used arginine substitutions to mimic the non-acetylated form as well as K to Q and K to E mutants to mimic acetylation and succinylation/malonylation at position 51, respectively. We first tried to generate a $\Delta efp \Delta efpL \Delta uup$ triple deletion in presence of any of the EfpL K51X variants but failed, thus indicating the importance of a lysine in that position. However, when testing these variants in $\Delta efp \Delta uup$ cells, we observed clear differences (Fig. 4C). In minimal medium with acetate as the sole carbon source only EfpL K51R supported significant growth, whereas the wild-type EfpL as well K51E failed, further substantiating the *in vitro* translation data, which shows inactivation of the protein by acylation. The little growth with the K51Q variant might be explained, that Q is rather reminiscent to lysine than acetyl-lysine regarding chain length.

*"We used E. coli cells that dependent on EfpL as the sole ribosome rescue system for stalls at XPX and XPPX ($\Delta efp \Delta uup$), and tested growth in acetate medium, expressing EfpL K51 substitutions (Fig. 4C). Arginine (K51R) was used to mimic the non-acetylated state, glutamine (K51Q) served as an acetyl-lysine mimic, and glutamate (K51E) introduced a negative charge similar to malonylation and succinylation. All variants were expressed from a low-copy number plasmid64 under the control of the native efpL promoter (PefpL). Under these conditions, only the K51R culture grew comparable to $\Delta efp \Delta uup$ cells ectopically expressing efp. By contrast, the culture with the K51Q variant turned only slightly turbid and we did not observe an increase in culture density for the K51E variant nor with wild-type EfpL. This, in turn, confirmed our previous assumptions, demonstrating that both chain length and charge at EfpL position 51 are crucial for protein activity. In conclusion, our combined *in vivo*, *in vitro*, and *in silico* data clearly demonstrate that EfpL is inactivated by acylation."*

5) In the authors' mass spectrometry data, have you ever detected acetylation at the K51 position? For example, in glucose media?

We would like to point out that several independent studies identified the multiple acylations we referring to (see Weinert et al. 2013, Kuhn et al. 2014, Qian et al. 2016). Additionally we performed immunoblotting analyses from endogenous EfpL grown either in LB or LB supplemented with 40mM Glucose, and detected acetylation only in the latter (Supplementary Fig. S4C,D).

Is there any chance acetylation affects not only activity but also protein stability?

When we produced our acetylated EfpL variants we did not observe any differences in yield. Moreover, we did not see any differences in protein levels when comparing acylation favoring and disfavoring conditions (Supplementary Fig. S4)

Similarly, authors predicted that in glucose media, acylation will occur due to changes in acetyl CoA levels, and a difference in growth rate under these conditions was shown. Does EFPL acetylation actually occur in cells grown in glucose media?

As stated under 5) we performed immunoblotting analyses from endogenous EfpL grown either in LB or LB supplemented with 40mM Glucose, and detected acetylation only in the latter (Supplementary Fig. S4C,D).

Is there any relative mRNA/protein/activity of EFPL under glucose conditions?

We did not see any changes in expression and protein levels Please see Supplementary Fig. S3D,E and S4B. Activity is completely abolished (compare Fig 2C with Fig. 3E and see Supplementary Fig. S3)

6) What can be concluded about the EFPL-dependent ribosome stalling motif? The conclusions of the motif analysis are unclear. For example, is there a common ribosome stalling motif among the 18 metabolic interconversion enzymes that are EFPL-dependent? How consistent is this with the efpL-dependent motif analyzed earlier?

Inclusion of uup mutant strains clearly shows, that EfpL is capable in resolving all stalls at any XP(P)X arrest motif (Fig. 2B & Supplementary Fig. S6). Nevertheless, the sequence logo based on translations influenced exclusively by EfpL, fit to the "preferred" rescue spectrum given in the new Extended Data Fig. 4C and Supplementary Tab. S3).

7) Since authors used gene deletion instead of a substitution mutant, can you really say that the lack of difference in growth between the wild-type and efpL mutant in glucose is due to EFPL K51 acetylation?

We agree that our previous *in vivo* data was rather indirectly pointing to an effect by acylation. Accordingly, and stated under 4) we now used arginine substitutions to mimic the non-acetylated form as well as K to Q and K to E mutants to mimic acetylation and succinylation/malonylation at position 51, respectively. We first tried to generate a $\Delta efp\Delta efpL\Delta uup$ triple deletion in presence of any of the EfpL K51X variants but failed, thus indicating the importance of a lysine in that position. However, when testing these variants in $\Delta efp\Delta uup$ cells, we observed clear differences. In minimal medium with acetate as the sole carbon source only EfpL K51R supported significant growth, further substantiating the *in vitro* translation data, which shows inactivation of the protein by acylation. The little growth with the K51Q variant might be explained, that Q is rather reminiscent to lysine than acetyl-lysine regarding chain length.

Coreviewer's comments:

The manuscript by Sieber et al. identifies new features of EF-P paralog based on *in silico* methods. The authors propose new sites where EfpL engages, beyond the well-known XPPX motifs, and describe how its activity is regulated in *E. coli*. They also showed that the guanosine in the first position of the E-site codon is important for EF-P and EfpL recognition. The extended but unmodified $\beta 3\Omega\beta 4$ loop, conserved in EfpL, interacts with mRNA and rescues the ribosome from XP(P)X stalling sequences. EfpL is involved in promoting faster bacterial growth and sensing the metabolic state to regulate translation.

The evidence presented strongly supports the proposed model, and the manuscript is generally well-written and understandable. However, there are still some comments and suggestions for improvement:

We are grateful to the reviewer for the judgment of our paper.

Major comments:

As EfpL and EF-P have overlapping functions, does the expression level of efpL increase in the Δefp strain?

We have no evidence for this as exemplified by our promoter activity assays (Supplementary Fig. S12).

Lines 314-316: Why were $\Delta efpL$ mutant strains used although acylation occurs in the K51 of EfpL?

We apologize for not making this clear earlier. Since EfpL is modified at multiple sites by distinct acylations all competing for the same lysines, we initially decided to use $\Delta efpL$ strains to demonstrate their overall impact under various conditions. We apologize for any confusion this caused. In the revised version, we have restructured the manuscript and included the respective analysis as Supplementary Fig. S3. We even extended the data set by performing a competition experiment in LB +/-Glc (Figs. 2C & 3E). Further we now included substitution variants, K to R to mimic the non-acetylated form, and K to Q and K to E mutants to mimic acetylation and succinylation/malonylation at position 51, respectively (Fig. 4C).

Figure 4C: Why does the doubling time of the Efp/EfpL double-deleted strain compensated with EfpL ($\Delta Efp\Delta efpL + EfpL$) decrease in glucose media even though EfpL is deactivated by the acylation?

EfpL acetylation is predominantly a non-enzymatic though site-specific modification mediated by acetyl-phosphate. Overproduction increases the population of unmodified protein as shown for other acylations (Yanagisawa *et al.* 2012) and thus EfpL can rescue the phenotype even in acetylation favoring conditions (now Supplementary Fig. S3B). By contrast, a mild increase in EfpL protein copy number (by expressing *efpL* from a low copy plasmid and utilizing the native *efpL* promoter) did not support growth in acetate medium (new Fig. 4C).

Also, as the doubling time of $\Delta efpL$ and $\Delta efp\Delta efpL$ strains decreases in glucose media, it seems there are some unpredicted effects on growth when *efpL* is deleted in the presence of glucose.

Any apparent differences between the $\Delta efpL$ and $\Delta efp\Delta efpL$ in both conditions (LB +/- Glc) are statistically not significant.

Minor comments:

Figures 2A and 4C: Adding growth curves will help clarify the differences between the strains (for example, in Fig 4G).

This is a really good suggestion. We included these growth curves as new supplementary figure S3A.

Line 741: There are two periods at the end of the sentence.

We have removed the additional period.

Reviewer #3 (Remarks to the Author):

"I co-reviewed this manuscript with one of the reviewers who provided the listed reports. This is part of the Nature Communications initiative to facilitate training in peer review and to provide appropriate recognition for Early Career Researchers who co-review manuscripts."

We very much appreciated the co-reviewer's comments.

Reviewer #4 (Remarks to the Author):

This manuscript describes the results of an extensive series of studies the authors have done to elucidate the structure, function, and physiological role of EfpL, a protein found in bacteria that mitigates the pausing that occurs when ribosomes translate sequences that include runs of prolines. EfpL turns out to be a variant of EF-P, which has a broadly similar function, but EfpL's functional properties are distinctively different. Their mms represents a significant step forward in our understanding of how the elongation phase of protein synthesis is managed in bacteria.

We appreciate the reviewer's judgement on our manuscript.

Specific Comments:

1. The Introduction is terse, and, quite surprisingly, it includes little, if any, mention of the prior literature on EfpL, a.k.a. YeiP. There is some, and it must be discussed in the Introduction.

We apologize and have now extended the introduction by adding the structural predictions made in the review by Mudry and Rodnina.

"However, to date the molecular function of EfpL remains enigmatic²⁴. Bioinformatic analyses based on AlphaFold predictions indicate that EF-P-like proteins have a three-domain structure similar to EF-P, but they only share about 30% sequence similarity. Across the three domains, the C-terminal OB-domain shows the highest similarity between the two proteins. This domain's primary role is to interact with the small ribosomal subunit and the anticodon stem loop of the P-site tRNA. Notably, both EF-P and EfpL contain a tyrosine and an arginine in position 183 and 186, respectively (according to E. coli EF-P numbering), which are close enough to form hydrogen bonds with A42 of the P-site tRNA and G1338 within helix h29 of the 16S rRNA10. By contrast, the key residues in the KOW domain of EF-P, as well as the residue involved in specific recognition of prolyl-tRNA in stalled ribosomes, are less conserved. This in turn suggest that EfpL's role in translation diverges from those of canonical EF-Ps. In the frame of this study, we solved the structure of E. coli EfpL (EfpL) and uncovered its role in translation of XP(P)X-containing proteins."

About EfpL's function we were only able to find a PhD-thesis that deals with the topic but there is no corresponding peer review publication and the only target they associated with EfpL is MgtL. However, we included this citation, at the point where we reinvestigated the assumption.

"An arrest spectrum extension beyond diprolines has only been reported for IF5A thus far^{37,39} although there are weak indications in the literature that EF-P and similarly EfpL might assist in synthesis of the XPX containing sequence of the leader peptide MgtL⁴⁰⁻⁴³, which we able to substantiate (Supplementary Fig. S9)."

Further literature indications on transcriptional regulation by CRP could not be supported by our newly included data. To avoid any confusion, the respective literature was also only cited, at the point in the manuscript, where we reinvestigated the hypothesis.

"Following indications from a global analysis, efpL expression is regulated by carbon catabolite repression (Supplementary Fig. S12)⁴⁹. It was predicted that PefpL is a class II cAMP response protein (CRP)-dependent promoter. However, the putative CRP binding site deviates significantly from the consensus motif of the regulator. Consequently, we reinvestigated the hypothesized regulation analogous to previous studies⁵⁰ but did not observe any measurable effect (Supplementary Fig. S12B, C)."

Similarly, we included and discussed post-transcriptional regulation of EF-P and EfpL mRNA in the respective chapter.

"Subsequently, we extended our dataset to include conditions such as nutrient availability, acetyl-phosphate levels, heat, cold, acidic and alkaline pH, as well as high and low osmolarity (Supplementary Fig. S12D,E). Under all tested conditions, the promoter activities of Pefp and PefpL maintained a constant ratio of 10:1. Our findings are also consistent with "The quantitative and condition-dependent Escherichia coli proteome"²⁹, which shows that the protein copy number patterns of EF-P and EfpL perfectly match and follow other ribosomal factors (Supplementary Fig. S12F). Accordingly, post-transcriptional control of the respective efp and efpL mRNAs is attributed to maintaining the balance in protein copy number between the two proteins⁵²⁻⁵⁵."

If we have overseen any important literature on the topic, we are happy to include it into the final version of the manuscript.

2. The crystallography done has some shortcomings. For one thing, I/σ at the limiting resolution is reported to be 2.55, which is high. Traditionally, crystallographers have included data out to $I/\sigma = 2$, and papers published in the last decade or so have made a strong case for including data out to $I/\sigma = 1$. By omitting these high resolution data (or not collecting them) the authors have left a lot of quality on the table. It is also the case that for their structure, R_{free} is much higher than R than it should be, which suggests that the structure reported is not fully refined. A bit more work might have yielded something better. However, some might argue that the structure obtained is good enough for the authors' purpose.

We thank Reviewer#4 for his comments on inclusion of data from high-resolution shells, and additional refinements to improve our structure.

While the underlying data reduction tool (POINTLESS) recommends a resolution estimate of 2.67 Å based on $I/\sigma(I)$ and $CC_{1/2}$, we have used higher-resolution shells up to 2.29 Å. The resolution cut-off of 2.29 Å was chosen based on the following three criteria: $R_{\text{pim}} \leq 0.6000$, $I/\sigma(I) \geq 2.00$, and $CC_{1/2} \geq 0.3000$ by autoPROC (auto-processing pipeline). It should be noted that going to higher resolution shells (e.g., 2.2 Å) resulted in a negative $I/\sigma(I)$ indicating higher contributions from noise. The dataset also suffers from anisotropic behavior with best diffraction along one principal axis and worst along the other two axes, making the selection of high-resolution cutoff problematic (Evans, P. R., & Murshudov, G. N. (2013), PMID: 23793146).

Indeed, we have to confess the structure had not been refined to the maximum possible. We have now performed additional refinement of the structure by further including resolution shells up to 2.27 Å with an $I/\sigma(I)$ of 2.0, as suggested. The R-values are now comparable to structures at similar resolution. We are thus grateful for this initiation. We have now replaced all panels in Supplementary figure 2 as well as panel 1D in the main text with the finalized structure and updated the text accordingly.

We have attached the PDB report for our updated crystal structure. Note that an ongoing server issue currently does not allow us to provide the official PDB report for review. We thus include the as an alternative the "not for review" version, which is identical in content, but derived from a standalone server with no bugs creating the output. It clearly shows the improved structural statistics. We hope this is sufficient for now as it should not have an influence on the judgement of the structure or the derived data/conclusions. For the sake of time we will highly appreciate your understanding, and of course will be happy to send the official report as soon as provided by the server.

3. Figures 1E and 3F show computed structures for the interactions between external loops of EfpL with the CCA end of tRNA and anticodon loops. They are unconvincing. Extended Data Fig. 8 includes images of the electron density for both loops. For both, it is of poor quality, as the density of external loops often is due to local disorder. The authors would be wise to omit both Fig. 1E and Fig. 3F from their mms, and delete the corresponding text. They don't add much to the argument the authors are trying to make, and they could be just plain, flat out wrong.

We agree that care needs to be taken in not to over-interpret the derivable molecular information from HADDOCK runs with respect to the atomistic details. We performed those docking runs (including energy minimization) in order to get an estimate and visualization of potential contacts that could form in a force field, albeit using an isolated system (which we are aware of). As stated in the Method section, the likely restrained environment of the Ribosome is difficult to include, but has led us to implement e.g. the relative orientation of proteins and RNAs.

Please note that Fig. 3F represents EF-P (not EfpL). For the HADDOCK runs, the protein was used as is positioned and resolved in the Ribosome structure (PDB 6ENU), and the same is true for the RNA trinucleotide. Thus, here we used a starting model as it exists prior to HADDOCKing for comparison with the mutated RNA trinucleotide. We confess that protein sidechain orientations remain merely modelled during the docking procedure. However, the idea of using HADDOCK in order to explicitly compare possible contact formation based on an experimental RNA-protein structure holds true in Fig. 3F. Nonetheless, we agree with the reviewer that a central main text figure panel may be too misleading and thus decided to remove this panel. We would still keep the corresponding extended data/SI figure in the manuscript and suggest rephrasing the underlying main text as is done now. With this, we think to keep it more as a supportive finding, which would be in line with the exp. data regarding the derived motifs presented in the main text Fig. 3.

For Figure 1E, we agree that the overall setup is less based on an experimental structure because it involves EfpL, for which no structure within the ribosome or at least with RNA is available. We are aware of the low electron density in the corresponding loop (now adjusted in Ext. Data Figure 2B), while we feel the loop backbone conformation is convincingly visible. Again, it is HADDOCK to suggest comparative contact formation (excluding the context), while allowing sufficient flexibility and movement between partners at the site of interaction (see details in Methods). As the derivable models are well-converging and still bear valuable information (e.g. testable through mutants) we would be happy to leave the data in, but have now removed the panel from the main text to Ext. Data Figure 3, based on the same argumentation as above. Hoping for accordance with this reviewer, we have adjusted the text and now more carefully include the docking and its outcome in that it supports a hypothesis regarding the link between loop length and central tip amino acid side chain lengths and modification.

Altogether, we hope to convince this reviewer that the modelling/docking data may add supportive value and/or provide a suggestion for interactions on the atomistic level, underlying the wet-lab experimental observations. To make the reader aware of the existing limitations of docking, we have now also added a sentence to the description of data in Fig. 1 in order to put the data into a realistic context.

4. There are two chains in the asymmetric unit of the crystals the authors solved. What is the relationship between the two? A cursory glance suggests that they are related by a non-crystallographic twofold. Which chain did the authors use for their structure/function analyses they did?

At concentrations tested for crystallization, the major population of the protein is monomeric (see included analytical size exclusion chromatography data below for you information). The asymmetric unit has two chains from the crystal packing. After model building and refinement, chain B was used based on the completeness and quality of the model according to the PDB validation report. The chains are highly similar showing an RMSD of 0.238 Å. Method section and figure legend (wherever applicable) now clearly state the chain we used as well as the HADDOCK run with EfpL.

Analytical size exclusion chromatography of EfpL at two different concentrations was performed on an Superdex75 Increase 10/300GL equilibrated with 20 mM Tris-HCl, 50 mM NaCl and 1 mM DTT at pH 8.0 using a Bio-Rad NGC FPLC. Flow rates were set to 0.75 mL/min. 100 μ L of EfpL at 50 μ M (yellow line) and 250 μ M (black line) concentrations were loaded. The run was monitored by measuring absorbance at 280 nm.

REVIEWERS' COMMENTS

Reviewer #1 (Remarks to the Author):

I am fully satisfied with the revision. Congratulations on a great story!

Reviewer #2 (Remarks to the Author):

The authors have appropriately responded to the reviewers' questions and approached the issues experimentally, which has made the content of the revised manuscript clearer. In particular, the complementation experiment with the efpL K substitutions in the efp uup double mutant background is very interesting, and I believe this experiment further strengthens the importance of efpL.

Co-reviewer's response

The revised manuscript has well reflected the comments which has been pointed out before. Also, supplemented data and manuscripts enriched their opinions which was helpful to fully understand this report.

Reviewer #3 (Remarks to the Author):

Reviewer #4 (Remarks to the Author):

I am satisfied by the responses of the authors to the points I raised when I reviewed the original version of this manuscript. I think their paper is useful contribution to the literature that deal with the protein factors that support protein synthesis in bacteria.